



# Early Eocene carbon isotope excursions in a lignite bearing succession at the southern edge of the proto-North Sea (Schöningen, Germany)

Olaf K. Lenz[1,2,] Mara Montag[2], Volker Wilde[1], Katharina Methner[3,5], Walter Riegel[1], Andreas Mulch[3,4]

[1]Senckenberg Research Institute and Natural History Museum Frankfurt, 60325 Frankfurt am Main, Germany.
[2]Institute of Applied Geosciences, Technical University Darmstadt, 64287 Darmstadt, Germany.
[3]Senckenberg Biodiversity and Climate Research Centre (SBiK-F), Senckenberganlage 25, 60325 Frankfurt am Main, Germany.
[4]Institute of Geosciences, Goethe University Frankfurt, 60438 Frankfurt am Main, Germany.
[5]Department of Earth System Science, Department of Geological Sciences, Stanford University, USA.

*Corresponding author:* Olaf K. Lenz (olaf.lenz@senckenberg.de)

**Abstract.** Situated at the southern edge of the proto-North Sea the lower Eocene Schöningen Formation of the Helmstedt Lignite Mining District, Lower Saxony, Germany is characterized by several lignite seams alternating with estuarine to brackish interbeds. Here, we present carbon isotope data of bulk organic matter ($\delta^{13}C_{TOC}$), organic carbon content (%TOC), and palynomorphs from a 98 m thick sequence of the Schöningen Formation embedded into a new robust age model. This is based on eustatic sea-level changes, biostratigraphy, and a correlation to existing radiometric ages. Based on the $\delta^{13}C_{TOC}$ data we observe six negative carbon isotope excursions (CIEs) reflecting massive short-term carbon cycle perturbations. A strong CIE of -2.6 ‰ in $\delta^{13}C_{TOC}$ values in the Main Seam and the succeeding marine interbed can be related to the Paleocene–Eocene Thermal Maximum (PETM). The subsequent CIE of -1.7 ‰ in $\delta^{13}C_{TOC}$ values may be correlated with the Eocene Thermal Maximum 2 (ETM2) or slightly older events preceding the ETM2. High-amplitude climate fluctuations including at least 4 minor CIEs with a maximum negative shift of -1.3 ‰ in $\delta^{13}C_{TOC}$ in the upper part of the studied section are characteristic for the EECO. Palynological analysis across the Main Seam proved that shifts in $\delta^{13}C_{TOC}$ values are correlated with changes in the peat forming wetland vegetation, specifically the change from a mixed angiosperm and gymnosperm flora to an angiosperm dominated vegetation at the onset of the PETM. The PETM-related CIE shows a distinct rebound to higher $\delta^{13}C_{TOC}$ values shortly after the onset of the CIE, which is here recognized as a common feature of terrestrial and marginal marine PETM-records worldwide and may be related to changes in the vegetation including different carbon isotope budgets of gymnosperms and angiosperms.



# 1 Introduction

The early Paleogene is characterized by a long-term warming trend that culminated in the Early Eocene Climatic Optimum (EECO; Zachos et al., 2001) between *c.* 53.3 and 49.1 Ma before present (Westerhold et al., 2018b). However, this long-term trend was interrupted by several short-term thermal events. These events were characterized by perturbations in the global carbon cycle, which are represented by negative carbon isotope excursions (CIE) caused by the rapid and massive input of $^{13}$C-depleted carbon into the atmosphere-ocean system (e.g., Kennett and Scott, 1991; Dickens, 2001; Cramer et al., 2003;

Lourens et al., 2005; Zachos et al,. 2003, 2008, 2010; Sluijs and Dickens, 2012; Kirtland Turner et al., 2014). The most prominent excursion was the Paleocene-Eocene Thermal Maximum (PETM) or Eocene Thermal Maximum 1 (ETM1), 55.93 Ma before present (e.g., Kennett and Stott, 1991; Bains et al., 2000; Röhl et al., 2000; Westerhold et al., 2017), characterized by a sudden temperature peak at the transition from the Paleocene to the Eocene that lasted for about $170 \pm 30$ kyr (Röhl et al., 2007; Westerhold et al., 2017; Zeebe and Lourens, 2019). During this event, the global mean surface temperature

(GMST) increased from 22 to 28 °C in the latest Paleocene to 27 to 35 °C (Inglis et al., 2020). Assuming a pre-industrial GMST of 14°C, this is about 17 °C warmer than pre-industrial (Inglis et al., 2020). This temperature increase led to a sea level rise of 3 to 5 m (Sluijs et al., 2008) and both, marine and continental ecosystems changed noticeably on a global scale (Gingerich, 2006; McInerney and Wing, 2011). However, the PETM was hardly an extinction event, but a "kick-starting event" with regard to evolution (Speijer et al., 2012), especially among vertebrates (e.g., mammals; Gingerich, 2006; Hooker

and Collinson, 2012) and plants (Wing et al., 2005; Jaramillo et al., 2010; McInerney and Wing, 2011).

Understanding the PETM is of high interest due to the potential to inform predictions for the effects of present and future global warming that can be expected due to the emission of greenhouse gases from the burning of fossil fuels (McCarren et al., 2008; Zachos et al., 2005; Zeebe et al., 2016). In particular, the long-term effects and responses of ecosystems on a kiloyear to millennial scale can be studied in marine as well as in terrestrial PETM records. However, there is still a need to

study the impact of global warming on the respective ecosystems, especially in well constrained high-resolution terrestrial records.

During the early Eocene the PETM was followed by a number of other short-term events (see Cramer et al., 2003; Westerhold et al., 2017). Among those the ETM2 (54.05 Ma before present; Lourens et al., 2005; Sluijs et al., 2009; Westerhold et al., 2017, 2018b) and the ETM3 (X- or K-event; 52.85 Ma before present; Cramer et al., 2003; Röhl et al.,

2005; Westerhold et al., 2017, 2018b) events are well pronounced in marine isotope records but did not reach the intensity of the PETM. However, their impact on environments and ecosystems are yet not well understood (Abels et al., 2016).

Meanwhile, more than 25 other short-term CIEs of minor importance have been recognized in marine records from the Atlantic (Sexton et al., 2011; Kirtland Turner et al., 2014) and the Pacific (Westerhold et al., 2018b) for the period between 56 and 45 Ma. At least four of them (H1, H2, I1, I2) were apparently orbitally controlled and tied to short eccentricity

maxima (Abels et al., 2016). Other short-term CIEs of the lower Eocene (including the PETM) show also a striking coincidence with short and long eccentricity cycles (Westerhold et al., 2018b; Zeebe and Lourens, 2019).



The long-term warming phase of the EECO included a whole series of the early Eocene CIE-related thermal events (Westerhold et al., 2018b) with its beginning placed at the J event at 53.62 Ma (Lauretano et al., 2015; Westerhold et al., 2018b). During the peak of the EECO the GMST was 10 to 16 °C above today but did not reach the GMST of the PETM

(Inglis et al., 2020). The succeeding onset of global cooling at 49.14 Ma led to a steady but slow decline in temperature until the Eocene-Oligocene boundary where the sudden shift from the Paleogene greenhouse system to the actual icehouse system occurred (Zachos et al., 2008; Westerhold et al., 2020). The middle and late Eocene cooling trend was interrupted only by few short-term warming events (Westerhold et al., 2018b), such as the Late Lutetian Thermal Maximum (LLTM) at 41.52 Ma (Westerhold et al., 2018a) and the Middle Eocene Climatic Optimum (MECO; Bohaty and Zachos, 2003) between 40.6

and 40.0 Ma (Bohaty et al., 2009; Mulch et al., 2015; Methner et al., 2016). Compared to the other Eocene thermal events the MECO does not show a prominent CIE in most of its records (Bohaty et al., 2009; Giorgioni et al., 2019) which, together with a relatively long duration of ~500 kyr, indicates that it was most probably triggered by other factors than the rapid release of (organic) carbon to the atmosphere (Giorgioni et al., 2019).

Despite high resolution records of the short- and long-term warming events of the Paleogene greenhouse from deep oceans,

the spatial effects of most of these hyperthermals are only poorly understood due to inadequate data coverage (Westerhold et al., 2018b). Especially, there are only few records providing insight into the response of terrestrial ecosystems. Long-term terrestrial carbon isotope records covering the Paleocene and the Lower Eocene are only available from the Bighorn Basin, Wyoming, USA (e.g., Gingerich, 2006; Abels et al., 2012, 2016). Other isotope records are confined to the PETM, such as those from some sections in China (Nanyang Basin; Zhu et al., 2010), Colombia (Jaramillo et al., 2010), or Europe (Cobham

Lignite, Great Britain; Collinson et al., 2003 and Vasterival, N-France; Storme et al., 2012).

The sedimentary succession of the former Helmstedt Lignite Mining District almost continuously covers the entire Paleogene greenhouse phase in the so-called Helmstedt Embayment at the southern edge of the proto-North Sea (Lenz, 2005; Riegel et al., 2012, 2015) (Fig. 1). This offers the rare opportunity to study Paleocene–Eocene near-coastal ecosystems which preserved the effects of long- and short-term climate perturbations across more than 10 million years.

Here we present a new and robust stratigraphic framework for the succession of the Helmstedt mining district at Schöningen based on biostratigraphy (dinoflagellate cysts) and comparison with the standard eustatic sea level curves. Carbon isotope analyses of bulk organic matter ($\delta^{13}C_{TOC}$) and organic carbon content (%TOC) from the lower part of the Schöningen Formation now significantly extend the existing carbon isotope stratigraphy (Methner et al., 2019) and allow to identify CIEs in the record that can now be correlated to known thermal events such as the PETM and the EECO. Palynological data from

the lowermost seam most probably including the PETM reveal a distinct response of the vegetation to the most significant thermal event in the Early Cenozoic.



## 2 Geological setting and stratigraphy

The Helmstedt Lignite Mining District is situated within the Paleogene Helmstedt Embayment (Fig. 1a) that represented the
mouth of the Mid-German Estuary (Standke, 2008b; Standke et al., 2010) opening towards the proto-North Sea between
uplifts corresponding to today's Harz Mountains in the South and the Flechtingen Rise in the North (Brandes et al., 2012).
At times, the estuary reached more than 100 km inland forming the so-called Leipzig Embayment (Blumenstengel and
Krutzsch, 2008; Standke, 2008a). Due to the interplay of eustatic changes in sea level, salt withdrawal from the subsurface,
and climate-related changes in runoff from the hinterland frequent changes between terrestrial and marine conditions
characterized the region and are now preserved in the unique succession of lignites and interbeds in the Paleogene of the
Helmstedt Lignite Mining District (Riegel et al., 2012; Wilde et al., 2021).

The Paleogene deposits of the Helmstedt Lignite Mining District are limited to two marginal synclines, which are separated
by a narrow core of Upper Permian (Zechstein) rocks belonging to the Helmstedt-Staßfurt salt-wall (Fig. 1b; Brandes et al.,
2012). A nearly 400 m thick succession in both synclines unconformably follows on Mesozoic sediments of Upper Triassic
and Lower Jurassic age (e.g., Stottmeister, 2007). In the western syncline the Paleogene sequence can be subdivided in the
Waseberg Formation (underlying sediments), the Schöningen Formation (Lower Seam Group), the Helmstedt Formation
(Upper Seam Group) and the overlying marine strata of the Annenberg-, Gehlberg-, and Silberberg Formations (Fig. 1c;
Gramann et al., 1975; Gürs, 2005; Riegel et al., 2012). The greensand of the Emmerstedt Formation between the Schöningen
and Helmstedt Formations is limited to the northwestern part of the syncline but missing at the Schöningen locality. Previous
age constraints for the succession were restricted to scattered biostratigraphic data from nannoplankton (Gramann et al.,
1975), dinocysts (Köthe, 2003), and palynomorphs (Pflug, 1952, 1986) together with some radiometric ages and dinocyst
occurrences from a core at Emmerstedt in the eastern syncline ca. 20 km north of Schöningen (Ahrendt et al., 1995). The
isolated data have been applied to both synclines simply by means of lithologic correlation suggesting a lower Eocene
(Ypresian) age for the Schöningen Formation and a middle Eocene (Lutetian) age for the Helmstedt Formation (Fig. 1c).

The Schöningen Formation as formerly exposed in the opencast mine Schöningen-Südfeld (Fig. 1c) of the western syncline
has a thickness of about 150 m, including nine laterally continuous lignite seams (Main Seam to Seam 9) and some
additional seams of limited extent, including "Seam L" and the "Sphagnum Seam" (Riegel et al., 2012; Riegel and Wilde,
2016). The Helmstedt Formation of the western syncline at Schöningen is composed of three lignite seams, the Lower Seam
("Unterflöz" or Seam 9; Riegel et al., 2012), Seam Viktoria, and Seam Treue (Fig. 1c).

It has previously been argued that the lower part of the Schöningen Formation at Schöningen covers the PETM since the
shallow marine deposits of Interbed 2 between Seams 1 and 2 (Fig. 1b) revealed a conspicuous peak in the abundance of
dinoflagellate cysts of the genus *Apectodinium* (Riegel et al., 2012). Since an *Apectodinium* acme with abundant
*Apectodinium augustum* (now *Axiodinium augustum*; Williams et al., 2015) is generally accepted as indicator for the PETM
in mid-latitudinal to high-latitudinal marine records (e.g., Bujak and Brinkhuis, 1998; Crouch et al., 2001; Heilmann-Clausen
et al., 1985; Iakovleva et al., 2001; Sluijs and Brinkhuis, 2009; Sluijs et al., 2007), a first study to locate the PETM-related



CIE at Schöningen concentrated on a section from Seam 1 to Seam 2 (Methner et al., 2019). The study indeed recorded a CIE ranging from the top of Seam 1 into the base of Seam 2, however, *Axiodinium augustum* is missing among the *Apectodinium* assemblage of Interbed 2. Therefore, the CIE could not unequivocally be related to the PETM and probably represents a later warming event of the lower Eocene.

## 3 Material and Methods

### 3.1 Studied section

During active mining in the western syncline of the Helmstedt Lignite Mining District excellent outcrops existed until 2016 in the now-abandoned opencast mines at Schöningen. Between 2005 and 2016 a comprehensive archive of more than 2000 samples has been acquired from more than 50 individual sections that is now hosted in the Senckenberg Forschungsinstitut und Naturmuseum Frankfurt. For the present study 211 samples from the former opencast mine Schöningen-Südfeld were selected for carbon isotope analyses in addition to the 121 samples of Methner et al. (2019). The isotope dataset now ranges from the base of the Main Seam up to the base of Interbed 7 (Fig. 1c) thereby covering the latest Paleocene and much of the lower Eocene. Due to accessibility in the active mine, the 98 m isotope record had to be composed from seven individual sections (for description see Supplementary Material S1), however, continuity was confirmed by lateral tracing of beds. For pollen analysis 105 samples of a section from the Main Seam in the adjacent former opencast mine Schöningen-Nordfeld were studied (for description of the section see Hammer-Schiemann, 1998). Cysts of dinoflagellates have been studied in four samples from Interbed 1, eight samples from Interbed 2 and two samples from the base of Interbed 3 in the former opencast mine Schöningen-Südfeld.

### 3.2 Carbon isotope analyses

Together with the 121 samples of Methner et al. (2019) a %TOC and $\delta^{13}C_{TOC}$ record of 332 samples with an average sample spacing of ~29.5 cm is now available for the lower 98 m of the Schöningen Formation at Schöningen. Sample preparation included freeze drying, grinding, removal of inorganic carbon (using 10% HCl for 24h at 40°C), centrifugation (4x at 2800 to 3000 rpm for 4 to 8 min) and sample drying (24h at 40°C). About ~0.2 mg of lignite and up to ~6 mg of clastics from the marine interbeds were analyzed with a Flash EA 1112 that was coupled to a ThermoFisher MAT 253 gas-source isotope ratio mass spectrometer at the Goethe University - Senckenberg BiK-F Joint Stable Isotope Facility (Frankfurt). An uncertainty of 0.2‰ was indicated for measured $\delta^{13}C_{TOC}$ values by analyzing USGS 24 and IAEA-CH-7 standard materials on a daily basis and replicate measurements of reference materials and samples. The total organic carbon concentrations [in %] have been calculated by relating the signal size of the samples and the averaged signal size of the daily standards (USGS 24, n =8). A typical error of ~0.5% based on mass spectrometric analysis, and maximum difference of ~7% in TOC contents of replicate measurements (including weighing uncertainties) was detected.



## 3.3 Palynological analyses

Palynological preparation of lignite and interbed samples followed standard procedures as described by Hammer-Schiemann (1998) including treatment with hot 30% hydrogen peroxide ($H_2O_2$) and 10% potassium hydroxide (KOH). Clastic interbed samples were treated with 38% hydrofluoric acid (HF) for several days. All samples were sieved through a 10 μm mesh sieve and the residues stored in glycerine. For light microscopy permanent glycerine jelly slides were produced. Remaining sample material, residues and slides are stored at the Senckenberg Forschungsinstitut und Naturmuseum Frankfurt.

To obtain a representative dataset for the Main Seam, at least 300 individual grains of pollen and spores were counted per sample at 400x magnification (raw data in Table S3). The palynomorphs were mainly identified based on the systematic–taxonomic studies of Thomson and Pflug (1953), Krutzsch (1970), Thiele-Pfeiffer (1988), Nickel (1996), Hammer-Schiemann (1998), and Lenz (2005). The simplified pollen diagram shows the abundance of the most important palynomorphs in percentages.

For biostratigraphy, dinoflagellate cysts have been studied qualitatively in the interbeds. Their identification is mainly based on Köthe (1990) and Gedl (2013), stratigraphic ranges are taken from Köthe and Piesker (2007).

## 4 Results and discussion

### 4.1 Age model

An accurate age model is a prerequisite for the identification of specific CIEs in the isotope record at Schöningen. Previous age assignments for the succession at Schöningen (Riegel et al., 2012; Brandes et al., 2012) were based on lithologic correlation with drill cores in the eastern syncline near Emmerstedt, ca. 20 km north of Schöningen, which were dated radiometrically and biostratigraphically (Ahrendt et al., 1995). K/Ar ages of glauconites are available for the middle part of the Schöningen Formation (52.8 ± 1.4 Ma) and for its top (49.9 ± 1.2 Ma) (Ahrendt et al., 1995). Transferring the age of 46.0 ± 1.2 Ma for the base of the Emmerstedt greensand (=Emmerstedt Formation; Ahrendt et al., 1995) to the Schöningen succession is hampered by the lack of greensand in an equivalent position. However, the interbed between Seam 8 at the top of the Schöningen Formation and Seam 9 at the base of the Helmstedt Formation can most probably be regarded as age equivalent (Riegel et al., 2012).

### 4.1.1 Dinoflagellates

Dinoflagellate cyst assemblages from marine horizons at Emmerstedt (Ahrendt et al., 1995) indicate that the Schöningen Formation can be assigned to the lower Eocene dinoflagellate zones D5b to D9na for northwestern Germany (Costa and Manum, 1988; Köthe, 1990, 2003). Cysts of dinoflagellates have also been recorded in interbeds with marine influence at Schöningen, but their provisional assignment to dinoflagellate zones by Riegel et al. (2012) is slightly modfied here.





The three interbeds 1, 2 and 3 from the lower part of the Schöningen Formation include dinoflagellate cyst assemblages of low diversity with a dominance of few species (Table 1, Supplementary Figs. S2 – S4).

    Interbed 1 is characterized by an essentially monogeneric assemblage of *Apectodinium* species with dominance of *A. homomorphum* but regular occurrences of *A. parvum*. Only single specimens of *Homotryblium* and *Glaphyrocysta* have been observed. On this basis Interbed 1 can be assigned to either zone D5 or the lower part of D6 (D6na). Zone D5 has been

subdivided into subzones D5na and D5nb (Köthe 1990) whereby subzone D5na has been characterized by the occurrence of *Axiodinium augustum* (formerly *Apectodinium augustum*), which is the diagnostic species for the PETM. However, since *A. augustum* is generally missing in marine sediments of northwestern Germany and has not been recorded at Schöningen, the age of Interbed 1 cannot be further constrained.

    Interbed 2 is characterized by a distinct acme of *Apectodinium* species, but includes a number of other taxa, e.g.

*Thalassiphora pelagica, Cordosphaeridium* sp.*, Glaphyrocysta* sp.*,* and a single specimen of the *Cleistosphaeridium placacanthum/ancyreum* group. Although the latter reaches the first peak in zone D7na, single specimens may appear as early as zone D5. Thus, based on the mass occurrence of *Apectodinium* Interbed 2 can best be assigned to zone D5 or the lower part of zone D6 (D6na). The lack of wetzelielloid cysts other than *Apectodinium,* i.e. *Wetzeliella* spp. and *Dracodinium* spp., which are diagnostic for zone D6, may be attributed to the estuarine environment at Schöningen, which did not include

the full range of marine dinoflagellates as known from the Eocene North Sea Basin. This precludes a precise delimitation of zones D5 and D6 and a comparison of the Schöningen assemblage with the Eocene North Sea dinocyst zonation of Bujak and Mudge (1994).

    Compared to interbeds 1 and 2 the base of Interbed 3 is characterized by a higher diversity of dinoflagellate cysts (Table 1, Fig. 2) with mainly long ranging species. Some of them have their first occurrence in the Paleocene or range into the

Oligocene or even Miocene (Fig. 2). The stratigraphic significance of the dinoflagellate cyst assemblage is therefore relatively low and only *Apectodinum homomorphum* as single representative of the genus limits the range of interbed 3 to an interval covering zones D7na to D9na. The lack of fully marine taxa is again striking. In contrast, cysts of dinoflagellate ecogroups indicating near-coastal shallow water areas with fluctuating salinities (Köthe 1990) such as *Thalassiphora* or *Homotryblium* dominate. Among the wetzelielloid cysts the genera *Deflandrea* or *Ceratiopsis*, which frequently occur in

Lower Eocene deposits of the southern North Sea Basin except for zone D7na (Köthe, 1990), are completely missing in Interbed 3. Since our evidence of dinocyst occurrences is, thus far, largely restricted to its base, this may be a hint that at least this part of Interbed 3 is within Zone D7na.

    In summary, Interbed 1 can be assigned to dinoflagellate zone D5nb, Interbed 2 includes zones D5nb/D6na and Interbed 3 zone D7na. This clearly indicates that the lower two interbeds are of lowermost Eocene age. Since zone D5na with the

dinocyst *Axiodinium augustum* diagnostic for the PETM cannot be identified in Interbed 1, the Paleocene/Eocene boundary is expected to be represented below within the Main Seam.



### 4.1.2 Eustatic sea level

Another approach to generating a stratigraphic framework for the Schöningen and Helmstedt Formations is provided by matching the alternation of lignites and marine interbeds of the Eocene succession at Schöningen with established sea level
curves (Haq et al., 1987a, b, 1988; Miller et al., 2005a; Kominz et al., 2008) for the period between 57 and 41 Ma (Fig. 3). Even if variations in the global sea level were modified at Schöningen by local and regional factors such as subsurface salt withdrawal and fluctuations of sedimentary input (Wilde et al., 2021), there is still a good correspondence between times of low sea level and peat accumulation, allowing for a relatively precise age assignment of the lignite seams (Fig. 3). Furthermore, these ages fit the K/Ar ages from the drill core at Emmerstedt, ca. 20 km N of Schöningen (Ahrendt et al.,
225   1995).

The lower three interbeds, Interbeds 1 to 3, were deposited in an estuarine system with tidally influenced estuarine channels. An upward increase in marine influence as recorded by the increasing diversity of the dinoflagellate cyst assemblages (Fig.2) indicates an overall deepening-upward trend representing a retrogradational system (Dalrymple and Choi, 2007; Osman et al., 2013). For the corresponding period between 56 and 53.5 Ma, all three sea level curves show a general increase of the sea
level and an overall transgressive trend, interrupted by short episodes of regression which can be matched with Seams 1 and 2 at Schöningen (Fig. 3). Therefore, an age of more than 53.5 Ma can be assumed for the three lower seams, which corresponds well with the lowermost Eocene age assigned to the dinocyst zones D5nb to D7na of Interbeds 1 to 3. Therefore, an age of ~56 Ma is suggested for the Main Seam which indicates that it most probably represents the transition from the Paleocene to the Eocene including the PETM.

For Seam 3 and the thin, locally developed Sphagnum Seam (Riegel and Wilde, 2016), there is no clear correspondence to the three sea-level curves (Fig. 3). This may be due to a short-term preponderance of regional and local factors or to higher-frequency sea-level changes not represented in the global records. The approximate position of this part of the succession at 53.1 to 52.4 Ma (Fig. 3) corresponds to the K/Ar glauconite age of 52.8 ± 1.4 Ma (Ahrendt et al., 1995) for the equivalent of Interbed 4 at Emmerstedt.

During the deposition of Interbed 4 the system was flooded by a major transgression that resulted in the so-called "Spurensand", a highly bioturbated clayey sand including occasional glauconite (Fig. 3; Lietzow and Ritzkowski, 2005; Riegel et al., 2012, Osman et al., 2013). Subsequent regression initiated peat formation of Seam 4 (Riegel et al., 2012). Although the global sea level curves generally show a high sea level during this time (Fig. 3), Miller et al. (2005a) and Kominz et al. (2008) suggest a slightly regressive tendency that could have led to a progression of the estuarine system. The
transgression at the time of the "Spurensand", which has been considered to corresponded to a maximum flooding surface and the end of a depositional cycle (Lietzow and Ritzkowski, 2005; Osman et al., 2013), is only pronounced in the curves of Haq et al. (1987a, b, 1988).

Various regression and transgression phases followed, which mostly led to the formation of thinner seams in the Schöningen Formation, e.g., Seam 4 (locally split into three partings), Seam 5, Seam L and Seam 7, each only a few to several decimeters





thick. Only Seam 6 is 4 m thick and matches an extended phase of low sea level (Fig. 3). According to the global sea level

variations seams 4 to 7 can be assigned to a period between 51 and 49 Ma (Fig. 3).

The lower Eocene part of the succession terminates with Seam 8. If the following Interbed 9 is regarded as equivalent to the

greensand of the Emmerstedt Formation further to the NW and in the eastern syncline, the K/Ar glauconite age of $46.0 \pm 1.1$

Ma (Ahrendt et al., 1995) from Emmerstedt implies that it already has a middle Eocene age. The change from the

Schöningen Formation to the Emmerstedt Formation is characterized by a significant hiatus at the top of Seam 8 (Osman et

al., 2013), which may be represented at Schöningen by a deep channel reaching down from slightly above Seam 8 to a level

well below Seam 7. In particular, the strong transgression at 47.6 Ma as indicated in the sea level curves (Haq et al., 1987a, b,

1988) may correlate to the onset of deposition of the marine Emmerstedt Greensand and its equivalents and thus limits the

age range for the deposition of Seam 8 from 48.4 to 47.6 Ma.

Three phases of a low sea level during the early middle Eocene at ~46.9 - 46.6 Ma, ~45.7 to 45.0 Ma, and ~43.0 to 41.9 Ma

easily correlate with the three seams of the Helmstedt Formation at Schöningen (Fig. 3). Especially the two long periods of a

low sea level may be responsible for the formation of the 10 m and 20 m thick Victoria and Treue seams, respectively, at the

top of the succession in the western syncline. This implies that the Helmstedt Formation does not include the MECO (40.6 to

40.0 Ma; Bohaty and Zachos, 2003) although paleobotanical and palynological data suggests a paratropical flora for the

respective part of the section (Riegel et al., 1999, 2012).

Overall, the lignite bearing terminal Paleocene to middle Eocene deposits in the western syncline have been assigned to a

2nd order transgressive succession (Osman et al., 2013), which is comprised of four 3$^{rd}$ order sequences and several higher

order sequences possibly related to Milankovitch cycles and characterized by stacked cycles of estuarine and marine sands

capped by coals. Maximum flooding surfaces occur at the top of Interbed 4, at the top of Seam 6, in the greensand of the

Emmerstedt Formation and at the top of the sediments underlying Seam 9 (Osman et al., 2013), which is in good

correspondence with the global sea level curves (Fig. 3).

## 4.2 CIEs in the basal Schöningen Formation

Clastic interbeds and lignites are clearly separated by their total organic carbon (TOC) content (Fig. 4), which is generally

above 43% in lignites reaching more than 60% in all seams except the thinner Seams 4, 5 and L (supplementary data Tab.

S1). TOC content in the brackish interbeds is generally below 10% and often below 1%. Only in transitions more than 30%

TOC may be reached.

Throughout the studied section, carbon isotope ratios of bulk organic matter range between -25.1 ‰ and -28.8 ‰ (Fig. 4). In

the Main Seam $\delta^{13}C_{TOC}$ varies between -25.4 ‰ and -28.7 ‰ (average $\delta^{13}C_{TOC}$ = -26.96 $\pm$ 0.85 ‰, n = 47) with the

comparatively high standard deviation (SD) caused by a great range of $\delta^{13}C_{TOC}$. In the lower 3.5 m of the Main Seam, the

values constantly fluctuate between -26.8 ‰ and -25.4 ‰. A significant decrease in $\delta^{13}C_{TOC}$ values within the succeeding 1.2

m indicates a first CIE in the record of the Schöningen Formation (onset of CIE 1). Between 4.57 m and 4.79 m an abrupt



drop to -28.0 ‰ can be observed followed by an interval with low values. $\delta^{13}C_{TOC}$ values briefly rise again to -25.7 ‰, before another sharp decline to -28.7 ‰ at the top of the Main Seam occurs (9.85 m, Fig. 4, Table 2).

Following the lowest value at the top of the Main Seam, the values in the lower 7 m of Interbed 1 show a continuous
recovery of ~2 ‰ to a $\delta^{13}C_{TOC}$ value of -26.8 ‰. Including recovery, CIE 1 therefore encompasses 3.53 m to 14.87 m of the studied section. Above, the $\delta^{13}C_{TOC}$ values decrease with strong fluctuations (Fig. 4). At 16.47 m and 18.35 m from the base of the section negative spikes of $\delta^{13}C_{TOC}$ with values below -28 ‰ have been measured while the overall lowest value of -28.8 ‰ is reached at 21.33 m. However, all of these negative $\delta^{13}C_{TOC}$ values are restricted to single samples. Following the lowest value $\delta^{13}C_{TOC}$ strongly increases by 3.1 ‰ to -25.7 ‰ at the top of Interbed 1 (Fig. 4).

$\delta^{13}C_{TOC}$ values from the top of Seam 1 to the middle of Seam 2 have been identified as a distinct CIE (Methner et al. 2019). Both, the abrupt decrease of $\delta^{13}C_{TOC}$ values at the top of Seam 1 and their gradual increase within the lower half of Seam 2 occur within lignites remote from marine influence although bracketing clastic Interbed 2 with its prominent *Apectodinium* acme.

Following the pronounced high in the upper half of Seam 2 $\delta^{13}C_{TOC}$ values decrease again to -27.8 ‰ within the lower 1.1 m
of Interbed 3 and remain largely constant with only slight fluctuations up to the base of Seam 4 at 76.8 m (average $\delta^{13}C_{TOC}$ = -27.3 ± 0.4 ‰, n = 53). Between 67.35 m and 68.73 m, however, there is a short, but rapid drop in Interbed 4 by 1.3 ‰ to -28.2 ‰, which we tentatively interpret as CIE 3 (Fig. 4, Table 2).

Within Seam 4, $\delta^{13}C_{TOC}$ values rise to -26.8 ‰ and to -26.1 ‰ in the lower 70 cm of Interbed 5 after an abrupt drop to -28.2 ‰ at the interface between the seam and the interbed. The following values in Interbed 5 remain constantly high (average
$\delta^{13}C_{TOC}$ = -26.4 ± 0.2 ‰, n = 9) and no major change is recorded for Seam 5 (average $\delta^{13}C_{TOC}$ = -26.6 ± 0.2 ‰, n = 6). In Interbed 6 and the subsequent Seam 6 further significant fluctuations in $\delta^{13}C_{TOC}$ values may be observed which should be regarded as minor CIEs. CIE 4 is identified from the top of Seam 5 into the base of Interbed 6 (82.89 to 85.99 m) where $\delta^{13}C_{TOC}$ drops by 1.0 ‰ to -27.7 ‰ before values rise again to previous levels (Fig. 4, Table 2). Within Interbed 6 and from the top of Interbed 6 up to the top of Seam 6 two small but still pronounced excursions of 0.8 ‰ (CIE 5, between 88.25 and
91.7 m) and 0.7 ‰ (CIE 6, between 94.35 and 97.51 m) occur (Fig. 4, Table 2) over several samples each. Furthermore, an overall trend towards somewhat higher $\delta^{13}C_{TOC}$ values can be recognized from the base of Interbed 6 to the base of Interbed 7 (Fig. 4).

### 4.3 Effects of changes in lithology and mixing of carbon sources on $\delta^{13}C_{TOC}$ values

When comparing the six observed negative shifts in $\delta^{13}C_{TOC}$ values with known CIEs it is necessary to differentiate between
shifts occurring within seams or interbeds and shifts occurring at lithologic boundaries (Fig. 4). In the Schöningen section organic matter was commonly mixed into the brackish interbeds from adjacent peat mires and other terrestrial sources which may have induced shifts in $\delta^{13}C_{TOC}$ values independent of climate perturbations. For example, the increase in $\delta^{13}C_{TOC}$ at the base of Interbed 2 and the decrease at its top correlate with changes in lithology (Fig. 5) while the shift to lower $\delta^{13}C_{TOC}$





values at the top of Seam 1 and the gradual increase back to higher values within Seam 2 both occur within an individual
lignite bed (Fig. 4).

The most important decrease of $\delta^{13}C_{TOC}$ values starts well within the Main Seam, 6 m below the top at 3.53 m, independent
of lithology (Fig. 4). The gradual shift to higher values in the lower part of Interbed 1 (Fig. 5) likewise takes place without a
change in lithology and may reflect the recovery phase of CIE 1. On the other hand, a similar increase at the transition from
Seam 1 to Interbed 2 (Figs. 4, 5; Methner et al., 2019) may be related to the change to the estuarine environment of Interbed
2 to which, however, a significant amount of terrestrial carbon has been contributed (Riegel et al., 2012).

Since marine organic matter is depleted in $^{13}C$ (Sluijs and Dickens, 2012) mixing of $^{13}C$ from marine and terrestrial sources
will influence fluctuations of $\delta^{13}C_{TOC}$ values. For instance, massive driftwood and occasional rooting have been observed in
the lower part of Interbed 1 (Riegel et al., 2012) while the scattered negative values in the upper half of Interbed 1 can be
related to a higher proportion of marine TOC. Furthermore, a significant $\delta^{13}C_{TOC}$ increase is associated with higher TOC
contents signaling the transition to the succeeding Seam 1.

A general decrease in $\delta^{13}C_{TOC}$ values can be observed in the younger interbeds above Seam 2 (Fig. 5), which show a greater
marine influence (Riegel et al., 2012; Osman et al., 2013). For instance, there is a significant decrease of more than 2 ‰ at
the transition to Interbed 3 followed by fluctuating but relatively low $\delta^{13}C_{TOC}$ values up into the lower half of Interbed 4.
That the $\delta^{13}C_{TOC}$ values for the intercalated Seam 3 remain within the range of the overall fluctuations may be due to the low
number of samples analyzed for this seam.

Following a distinct single-sample decline to a value below -28 ‰ in the middle of Interbed 4 (CIE 3), comparatively low
$\delta^{13}C_{TOC}$ values persist in the upper part of Interbed 4. With the return to terrestrial conditions the values increase
significantly in the succeeding Seam 4 (Fig. 5). The strong drop of the $\delta^{13}C_{TOC}$ values at the top of Seam 4 again may be
related to the change in facies coupled to an increasing contribution of $^{13}C$ depleted marine organic matter (Fig. 5). However,
in the succeeding lower 30 cm of Interbed 5 the values strongly increase which is comparable to the situation in interbeds 1
and 2 and may indicate that the decline of the $\delta^{13}C_{TOC}$ values at the transition between Seam 4 and Interbed 5 is related to
changes in lithology.

At the base of Seam 5 the values are similarly high as in the underlying Interbed 5. However, a significant decrease in
$\delta^{13}C_{TOC}$ values can be observed from the base to the top of the seam (Fig. 5) where it continues into the base of Interbed 6
which may have been enforced by the transition to marine deposits with generally lower $\delta^{13}C_{TOC}$ values. However, in spite of
this effect, the part of the section from the base of Seam 5 into the lower part of Interbed 6 has been recognized as CIE 4,
because $\delta^{13}C_{TOC}$ values gradually and continuously rise again afterwards. A weak CIE has also been identified within Seam 6,
where it is independent of lithological changes (Fig. 5).

### 4.4 Thermal events in the Lower Eocene at Schöningen

Revision of the dinoflagellate zonation (Costa and Manum, 1988; Köthe, 1990, 2003) and comparison with eustatic sea level
changes (Haq et al., 1987a, b; Miller et al., 2005a; Kominz et al., 2008) in addition to the consideration of sequence



stratigraphic concepts (Osman et al., 2013) provide a robust stratigraphic framework for the interpretation of the $\delta^{13}C_{TOC}$ record from the lower part of the Schöningen Formation and the assignment of observed CIEs to known thermal events in the Lower Eocene.

### 4.4.1 CIE 1 (PETM)

Methner et al. (2019) already suggested that both the PETM and the associated Paleocene/Eocene boundary should be placed below Seam 1. The new $\delta^{13}C_{TOC}$ data now show a well-expressed CIE ranging from the Main Seam into the Interbed 1 (CIE 1, Figs. 4, 5). With a total drop of 3.3 ‰ in $\delta^{13}C_{TOC}$ values respectively 2.6 ‰ (CIE magnitude; Table 2) it is the strongest excursion in the entire section, thus far. Since Interbed 1 has now been placed in the earliest Eocene dinoflagellate zone

D5nb and the eustatic sea-level curve suggests an age of *c*. 56 Ma for the Main Seam (see above) CIE 1 most probably represents the PETM. As a consequence, we interpret the Paleocene/Eocene boundary to be located within the Main Seam. Termination and recovery phase of CIE 1 in the lower part of Interbed 1 were probably coupled to a sea level rise due to thermal expansion of the seawater (Sluijs et al., 2008) or melting of small ephemeral ice sheets in Antarctica (Speijer and Wagner, 2002; DeConto and Pollard, 2003; Miller et al., 2005b; Hollis et al., 2009) as a direct consequence of the PETM.

Sluijs et al. (2008) already suggested a link between the global sea level and hyperthermal events and recognized an eustatic sea level rise for the North Sea Basin during the PETM and the ETM2. A rise of sea-level is manifested in the Schöningen Formation by the change from the peat forming phase of the Main Seam to the brackish deposits of Interbed 1. Similar transitions from terrestrial to brackish deposits in conjunction with the PETM at the southern edge of the North Sea Basin were also reported from other sites, such as Cobham in southeastern England (Collinson et al., 2003) and the Oise region in

France (Cavagnetto, 2000; Nel et al., 1999; Quesnel et al., 2014). Therefore, our carbon isotope record is consistent with peat formation during the onset of the PETM and widespread inundation of coastal lowlands due to sea-level rise during the later stage of the PETM (see e.g., Denison, 2021).

The lack of the key species *A. augustum* within the dinoflagellate assemblage of Interbed 1 is not uncommon for PETM deposits of the southern North Sea Basin. In the French Dieppe and Paris basins the PETM-related CIEs similarly started

with terrestrial and coastal deposits and continued into lagoonal and shallow marine sediments, which are also characterized by an extremely pronounced *Apectodinium* acme without *A. augustum* (Iakovleva et al., 2013). In the marginal marine to lagoonal Tienen Formation in Belgium *A. augustum* is only known from the lower part of the PETM interval and missing above (Steuerbaut et al., 2000). Therefore, *A. augustum* is generally sparse and restricted to the lower part of the PETM interval in the southern North Sea Basin (Steurbaut et al., 2000). Since the lower part of the PETM interval in the

Schöningen Formation is included within the Main Seam *A. augustum* cannot be expected in Interbed 1.

The negative shift of -3.3‰ in $\delta^{13}C$ in the Main Seam is smaller than the shift of -4.7 ± 1.5 ‰ known for the PETM in other terrestrial records, but larger than the shift of -2.8 ± 1.3 ‰ in marine deposits (McInerney and Wing, 2011). The recovery phase in the lower part of Interbed 1 spans a positive shift of only about 2 ‰, which is roughly within the range known from other marine records, but overall $\delta^{13}C_{TOC}$ values do not return to pre-CIE values attained in the lower part of the Main Seam.



Most likely, admixture of marine organic matter depleted in $^{13}$C prevented the full recovery to $\delta^{13}C_{TOC}$ values as high as those recorded in the purely terrestrial organic matter of the Main Seam. However, the shift from the negative peak of $\delta^{13}C_{TOC}$ values at the top of the Main Seam to increasing values in the interbed at or near the lithological boundary has to be treated with caution since the PETM driven transgression in such a coastal setting may have caused a hiatus at the top of the Main Seam (Denison, 2021) and obscured an even more negative shift of $\delta^{13}C_{TOC}$ values.

The onset of a CIE is generally regarded as the time between the pre-CIE carbon isotope composition and the most depleted values (McInerney and Wing, 2011). Compared with the sharp drop commonly observed in the onsets of marine records (e.g., Röhl et al., 2007; Westerhold et al., 2018b; Fig. 6), the decrease in $\delta^{13}C$ values appears to be more gradual in the Main Seam. However, based on an average compaction rate of 3:1 (Widera, 2015) and an average rate of deposition for tropical peat of 3 mm/year (Stach et al., 1982), one meter of lignite in the Schöningen Formation may represent about 1.000 years. Therefore,

the 6.50 m of lignite representing the onset in the Main Seam may cover *c.* 6.5 kyr. This is well in accordance with a time of less than 10 kyr that has generally been estimated for the onset of the PETM-related CIE (Zachos et al., 2005).

Methner et al. (2019) initially noted close similarities between the $\delta^{13}C_{TOC}$ values from CIE 2 of the Schöningen section and those from other peat mires at the southern coast of the North Sea assigned to the PETM, e.g. the Cobham Lignite in the UK (Collinson et al., 2003, 2009; Pancost et al., 2007) and the Vasterival section in France (Garel et al., 2013; Storme et al.,

2012). However, these similarities also exist with CIE 1, i.e. a similar -3.2 ‰ shift of $\delta^{13}C_{TOC}$, similar minimum $\delta^{13}C_{TOC}$ value of -27.5 ‰ to -28.8 ‰ and similar CIE magnitudes. Interestingly, the CIE magnitude that is calculated as the difference between the mean pre-CIE and the mean CIE values are slightly smaller than those of CIE 1 (-1.4 ‰; Table 2) in both, the Cobham Lignite and the Vasterival section (-1.6 ‰ resp. -1.5 ‰). If the CIE magnitude, however, is calculated as the difference between the mean pre-CIE values and the most negative value during the CIE (McInerney and Wing, 2011)

the value for the Schöningen CIE 1 (-2.6 ‰) is larger than that of the Cobham Lignite and the Vasterival section (-2.0 ‰ resp. -2.3 ‰). Nevertheless, the similarity in all of these characteristics supports the hypothesis that the wetlands in midlatitudinal European near-coastal environments reacted similar during latest Paleocene to early Eocene thermal events (Methner et al., 2019).

Since within-seam carbon sources are likely to be purely terrestrial, the fluctuations of $\delta^{13}C_{TOC}$ values in the CIE of the Main

Seam including a clear rebound to higher values are rather striking. Similar to fluctuations described from the Lower Eocene of the Dieppe-Hampshire Basin in northern France (Garel et al., 2020), the short-term return to pre-CIE $\delta^{13}C_{TOC}$ values at around 2 m below the top of the Main Seam (Fig. 4) can only be explained by environmental factors or changes in vegetation. Especially pronounced is the change in pollen assemblages at the beginning of the CIE. In the lower part of the Main Seam they are characterized by abundant *Inaperturopollenites* averaging 18.4% of the total assemblage and frequent

*Nyssapollenites* probably representing a *Nyssa-Taxodium* swamp forest which has already been invoked for the near-coastal vegetation of Seam 1 (Lenz et al., 2021). In this part of the Main Seam $\delta^{13}C_{TOC}$ values fluctuate constantly between -26.8 ‰ and -25.4 ‰ (Fig. 7). However, with the change to an angiosperm forest dominated by Myricaceae (*Triporopollenites robustus/rhenanus*) at the beginning of CIE 1 $\delta^{13}C_{TOC}$ values suddenly drop to -28.0 ‰. With the occurrence of



*Pompeckjioidaepollenites sybhercynicus* (unknown botanical affinity) and *Sphagnum*-type spores values rise briefly to 25.7 ‰

(Fig. 7), but decrease again together with *P. subhercynicus* and *Sphagnum*-type spores at the top of the Main Seam. Thus, a clear correlation exists between intra-PETM fluctuations in $\delta^{13}C_{TOC}$ values and vegetation change.

A comparable shift from a mixed angiosperm-conifer flora in the Paleocene to an angiosperm flora at the onset of the PETM-related CIE was already reported for the Bighorn Basin in Wyoming (Smith et al., 2007, Diefendorf et al., 2010). Diefendorf et al. (2010) thereby suggested that a 4.6 ‰ decline in $\delta^{13}C$ values of atmospheric $CO_2$ at the onset of the PETM

can be attributed to the different carbon isotope budget of gymnosperm and angiosperm leaves. Thus, the rebound of $\delta^{13}C$ commonly observed the in terrestrial records (Fig. 6) should better be assigned at least at the Schöningen section to changes in vegetation rather than to reworking of organic matter as suggested for the Bighorn Basin, USA (Baczynski et al., 2013; Bataille et al., 2013) or the Nanyang Basin, central China (Chen et al., 2014).

### 4.4.2 CIE 2 (ETM2?)

Methner et al. (2019) suggested that the two negative excursions at the transition of Seam 1 to Interbed 2 and in the lower part of Seam 2 represent a single CIE (CIE 2 of the present paper) embracing a pronounced *Apectodinium* acme with mass occurrences of different species of *Apectodinium* in Interbed 2 and discussed a possible relation to the PETM. However, this excursion (-1.3 ‰ mean decrease in $\delta^{13}C_{TOC}$, Table 2) is smaller than CIE 1 in the Main Seam and other CIEs associated with the PETM in the terrestrial realm (McInerney and Wing, 2011), thus, implying another post-PETM hyperthermal

(Methner et al., 2019).

We now assign Interbed 2 with the distinct *Apectodinium* acme (Riegel et al., 2012) to the dinocyst zones D5nb or D6 (Fig. 2). Correlation with the eustatic sea level fluctuations limits deposition of Seam 1 and Seam 2 with Interbed 2 to a time between 53.7 and 55.1 Ma (Fig. 8). This includes the ETM2 or H1 event at 54.05 Ma which did not reach the intensity of the PETM (Westerhold et al., 2017) and is therefore a good candidate for CIE 2. According to Sluijs et al. (2008) the ETM2 was

also associated with a transgression in the North Sea basin which could well correlate to the change from terrestrial to marine deposits at the top of Seam 1.

However, CIE 2 can still not be unequivocally correlated (see Methner et al., 2019). According to Cramer et al. (2003) and Westerhold et al. (2017, 2018b) some minor CIEs such as the E1/E2, F and G events can be recognized in marine records between the PETM and the ETM2 events (Fig. 8). Some negative $\delta^{13}C$ excursions in a terrestrial to shallow marine lower

Eocene section in the Dieppe-Hampshire Basin in Normandy (France) were putatively linked to these events (Garel et al., 2020). Such a comparison may also work for CIE 2 in the Schöningen Formation, especially for the F (55.17 Ma, Westerhold et al., 2017) and the G (54.99 Ma, Cramer et al., 2003; Westerhold et al., 2017) events with ages approximating the age of Seam 1 (Fig. 8).





### 4.4.3 CIEs 3 – 6 (EECO)

The interval between Seam 3 and Seam 6 can be assigned to a period between 53 and 50 Ma (Fig. 8) that corresponds well to the EECO (53.26 - 49.14 Ma) as described from marine records (Westerhold et al., 2018b). According to the age model presented here, the terminal cooling phase of the EECO is not represented in the present dataset since it is probably situated above Seam 6.

The onset of the EECO, which is placed at the J event at 53.62 Ma (Westerhold et al., 2018b; Fig. 8), is difficult to locate in

the present $\delta^{13}C_{TOC}$ data from the Schöningen Formation, since the values remain low in Interbed 3 and do not show significant fluctuations (Fig. 4). However, correlation with the eustatic sea level (Fig. 2) dates Seam 3 to an interval between 52.5 and 53.1 Ma, a negligible difference to the 53.5 Ma suggested by Brandes et al. (2012) since peat accumulation of Seam 3 in any case started after the J event.

The phase of high amplitude isotope fluctuations between Seams 4 and 6 (Fig. 8) is characterized by highly varying $\delta^{13}C_{TOC}$

values including subordinate CIEs, which followed a phase of fairly stable values in Interbeds 3 and 4. However, within this interval none of the numerous minor CIEs generally characterizing the EECO (Westerhold et al., 2017; Fig. 8) can currently be assigned to any specific CIE in the Schöningen section. In general, the carbon isotope data from Schöningen reflect the global pattern of $\delta^{13}C$ data for this part of the EECO. This is further supported by the tentative correlation of a K/Ar age of 52.8 ± 1.4 Ma from the Emmerstedt core (Ahrendt et al., 1995) to the part of our section at Schöningen between Seam 3 and

Seam 6 (Riegel et al., 2012, Fig. 8).

### 5 Summary and conclusions

Bulk organic carbon isotopic data from a 98 m thick succession of alternating lignites and clastic deposits in the lower part of the Schöningen Formation at its type locality in the former Helmstedt Lignite Mining District, northern Germany, show at least six negative CIEs. New age constraints based on a biostratigraphical analysis of dinoflagellate cysts in combination

with a correlation of peat accumulation phases to lowstands of global sea level allow for a robust correlation of most of the recognized CIEs to known long- and short-term hyperthermals of the early Eocene:

(1) A strong CIE of -2.6 ‰ in $\delta^{13}C_{TOC}$ values in the upper part of the Main Seam (CIE 1) and the lower part of Interbed 1 can be related to the PETM. The onset of CIE 1 is restricted to the Main Seam and the minimum in $\delta^{13}C_{TOC}$ values was reached within less than 10 kyr. The sudden change from a *Nyssa-Taxodium* swamp forest to a largely

angiosperm-dominated flora in the peat forming vegetation strictly followed changes in $\delta^{13}C_{TOC}$ values suggesting that the CIE-related thermal event influenced the vegetation composition. A sudden change from a mixed gymnosperm-angiosperm association to an angiosperm-dominated flora occurs at the onset of the CIE. The lower part of the CIE in the Schöningen Formation occurs within purely terrestrial deposits. The PETM related transgression caused a change to brackish estuarine deposits. Following a possible hiatus at the lithological

boundary between seam and interbed the recovery phase of the CIE can be identified in the lower part of Interbed 1.



(2) CIE 2 has a negative shift of -1.7 ‰ in $\delta^{13}C_{TOC}$. It is comparable to the PETM-related CIE in the Main Seam/Interbed 1. Similar to CIE 1 it starts in the terrestrial environment of Seam 1 but is mainly included in the marine Interbed 2 and the lower part of Seam 2. Here, a transgression related to a thermal event has flooded coastal wetlands. This CIE can be tentatively correlated to the ETM2 event, but an assignment to other post-PETM CIEs
480         such as the F or G events preceding the ETM2 cannot be excluded.

(3) Between Seam 3 and Seam 6 of the Schöningen Formation at least 4 minor CIEs with a maximum negative shift of -1.3 ‰ in $\delta^{13}C_{TOC}$ values occur. Although it is impossible to assign these CIEs to specific events that are known from the marine record, the carbon isotope trend in this part of the studied section can clearly be correlated to the gradual long-term warming of the EECO. Especially the strongly fluctuating $\delta^{13}C_{TOC}$ values between Seam 4 and
485         Seam 6 point to high amplitude climate fluctuations and unstable climate conditions characteristic for the EECO shortly before the climate system shifted back to another cooling phase not covered by the presently studied section.

(4) Similar to other terrestrial PETM records $\delta^{13}C_{TOC}$ of the PETM at Schöningen within the Main Seam shows a characteristic rebound structure. Reworking of older organic material which was held responsible in other sections is almost excluded within our seam. However, the pollen and spore record clearly follows the isotope record which
490         makes changes in the vegetation a more probable cause.

**Data availability**

All shown and discussed data are available in the Supplement.

**Sample availability**

Samples are stored in the Senckenberg collections and are available upon request.

**Supplement**

The supplement related to this article is available online at: xxx

**Author contributions**

OL and VW designed the study. OL, MM and KM conducted the geochemical analyses and evaluated the results, WR and VW contributed field data and WR provided palynological data of the Main Seam. WR and OL were responsible for
biostratigraphy based on dinoflagellate cysts. OL wrote the initial draft of the manuscript and was responsible for the visualization of the results. All authors contributed to the interpretations and conclusions presented and edited the final version of this paper.



**Competing interests**

The authors declare that they have no conflict of interest.

**Acknowledgements**

Olaf Lenz acknowledges support through funding by the Deutsche Forschungsgemeinschaft (DFG, grants LE 2376/4-1, LE 2376/4-2). Katharina Methner acknowledges support through the Alexander von Humboldt Foundation. The authors thank Karin Schmidt for valuable field support as well as Jens Fiebig and Ulrich Treffert for technical assistance. We are also grateful to the Helmstedt Revier GmbH (formerly BKB and later EoN) for access to the sections and technical support in the 510 field.

**Financial support**

This research has been supported by the Deutsche Forschungsgemeinschaft (grants no. LE 2376/4-1, LE 2376/4-2) and the Alexander von Humboldt Foundation. The publication of this article was funded by the Open Access Fund of the Leibniz Association.

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





**Table 1: List of identified dinoflagellate cyst taxa in Interbed 1, Interbed 2 and the base of Interbed 3 of the Schöningen Formation in the western syncline of the Helmstedt Mining District.**

| Dinoflagellate species | IB 1 | IB 2 | IB 3 |
|---|---|---|---|
| *Achomosphaera crassipellis* (DEFLANDRE & COOKSON 1955) STOVER & EVITT 1978 | | | X |
| *Apectodinium homomorphum* (DEFLANDRE & COOKSON 1955) LENTIN & WILLIAMS 1977 | X | X | X |
| *Apectodinium longispinosum* (WILSON 1968) BUJAK & DAVEY 1983 | | X | |
| *Apectodinium parvum* (ALBERTI 1961) LENTIN & WILLIAMS 1977 | X | X | |
| *Apectodinium quniquelatum*(WILLIAMS & DOWNEY 1966) COSTA & DOWNIE 1979 | X | | |
| *Cleistosphaeridium placacanthum/ancyreum* group | | X | X |
| *Cordosphaeridium fibrospinosum* DAVEY & WILLIAMS 1966 | | X | X |
| *Cordosphaeridium gracile* (EISENACK 1954) DAVEY & WILLIAMS 1966 | | X | |
| *Cribroperidinium tenuitabulatum* (GERLACH 1961) HELENES 1984 | | X | |
| *Glaphyrocysta ordinata* (WILLIAMS & DOWNEY 1966) STOVER & EVITT 1978 | | | X |
| *Homotryblium tenuispinosum* DAVEY & WILLIAMS 1966 | | | X |
| *Hystrichokolpoma cinctum* KLUMPP 1953 | | | X |
| *Hystrichokolpoma rigaudiae* DEFLANDRE & COOKSON 1955 | | | X |
| *Phthanoperidinium sp.* | | | X |
| *Spiniferites ramosus* (EHRENBERG 1837) MANTELL 1854 | | X | |
| *Thalassiphora delicata* WILLIAMS & DOWNEY 1966 | | | X |
| *Thalassiphora pelagica* (EISENACK 1954) EISENACK & GOCHT 1960 | | X | X |





**Table 2: CIEs in the $\delta^{13}C_{TOC}$ dataset of the Schöningen Formation (Main Seam – Seam 6, western syncline).**

| CIEs in the Schöningen Formation | Min. and max. $\delta^{13}C_{TOC}$ values of each CIE (‰) | $\delta^{13}C_{TOC}$ range (‰)[a] | CIE onset (‰)[b] | CIE magnitude ("mean – mean", ‰)[C] | CIE magnitude ("mean – most negative", ‰)[d] |
|---|---|---|---|---|---|
| CIE 6 | -26.27 to -27.01 | 0.7 | -0.1 | -0.1 | -0.2 |
| CIE 5 | -26.60 to -27.43 | 0.8 | -0.3 | -0.3 | -0.5 |
| CIE 4 | -26.76 to -27.72 | 1.0 | -0.3 | -0.9 | -1.2 |
| CIE 3 | -26.88 to -28.20 | 1.3 | -0.7 | -0.7 | -1.1 |
| CIE 2 | -26.60 to -28.30 | 1.7 | -1.7[e] | -1.3 [e] | -1.5 [e] |
| CIE 1 | -25.40 to -28.67 | 3.3 | -0.60 | -1.4 | -2.6 |

[a]CIE range is calculated by the difference between the minimum and maximum value in the CIE. [b]CIE onset is calculated as the difference between the last pre-CIE and the first CIE sample. [c]CIE magnitude calculated as the difference between the mean pre-CIE and the mean CIE value. [d]CIE magnitude calculated as the difference between the mean pre-CIE and the most negative value of the CIE (following McInerney and Wing, 2011). [e]taken from Methner et al. (2019)








**Figure 1: (a) Paleogeographic map of northwestern Europe during the early and middle Eocene (adapted from Ziegler, 1990) showing the area of the Helmstedt Mining District ('H') at the southern coast of the Proto-North Sea. (b) Cross-section through the study area, showing the Helmstedt-Staßfurt salt wall and related synclines (modified after Brandes et al., 2012). (c) Generalized scheme of the Paleogene succession in the western syncline of the Helmstedt Mining District (adapted from Brandes et al., 2012, Lenz et al., 2021). The blue and red lines indicate the stratigraphic position of the studied sections.**





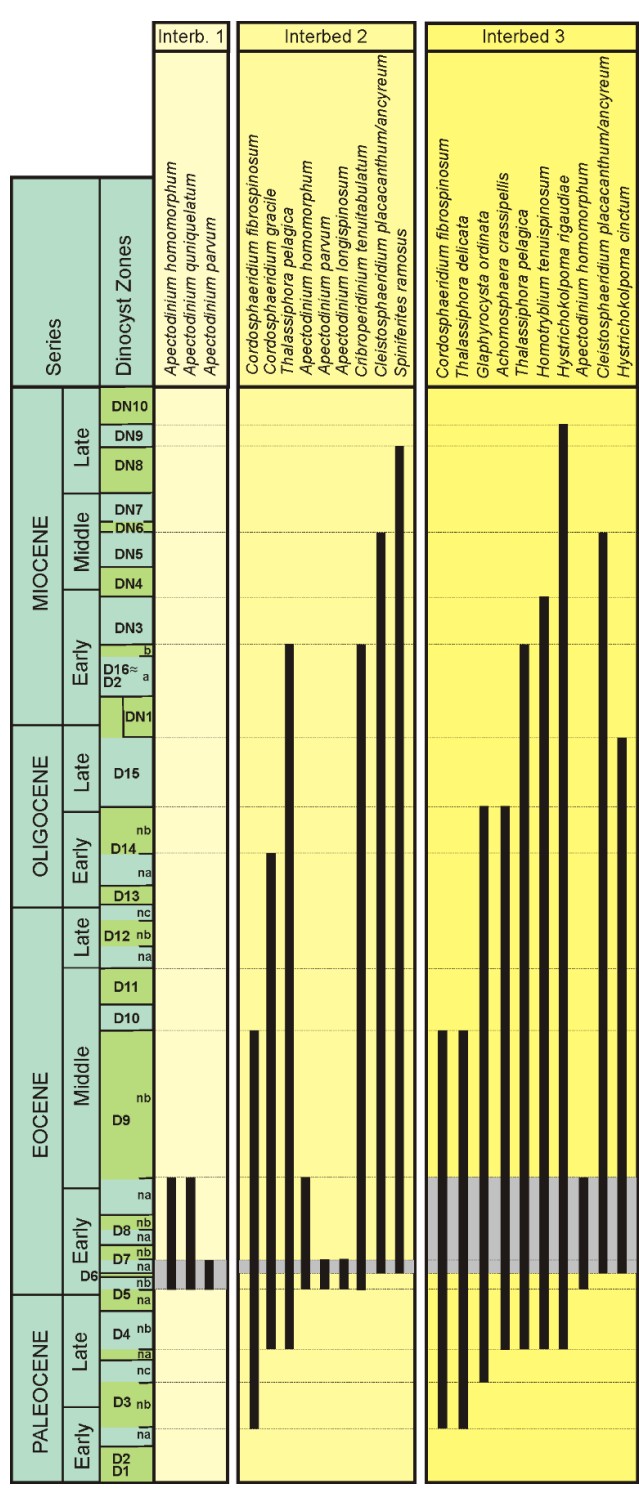

**Figure 2: Stratigraphic range of dinoflagellates identified in Interbed 1, Interbed 2 and at the base of Interbed 3 using the dinoflagellate zones of Köthe (2003) and Köthe and Piesker (2007).**





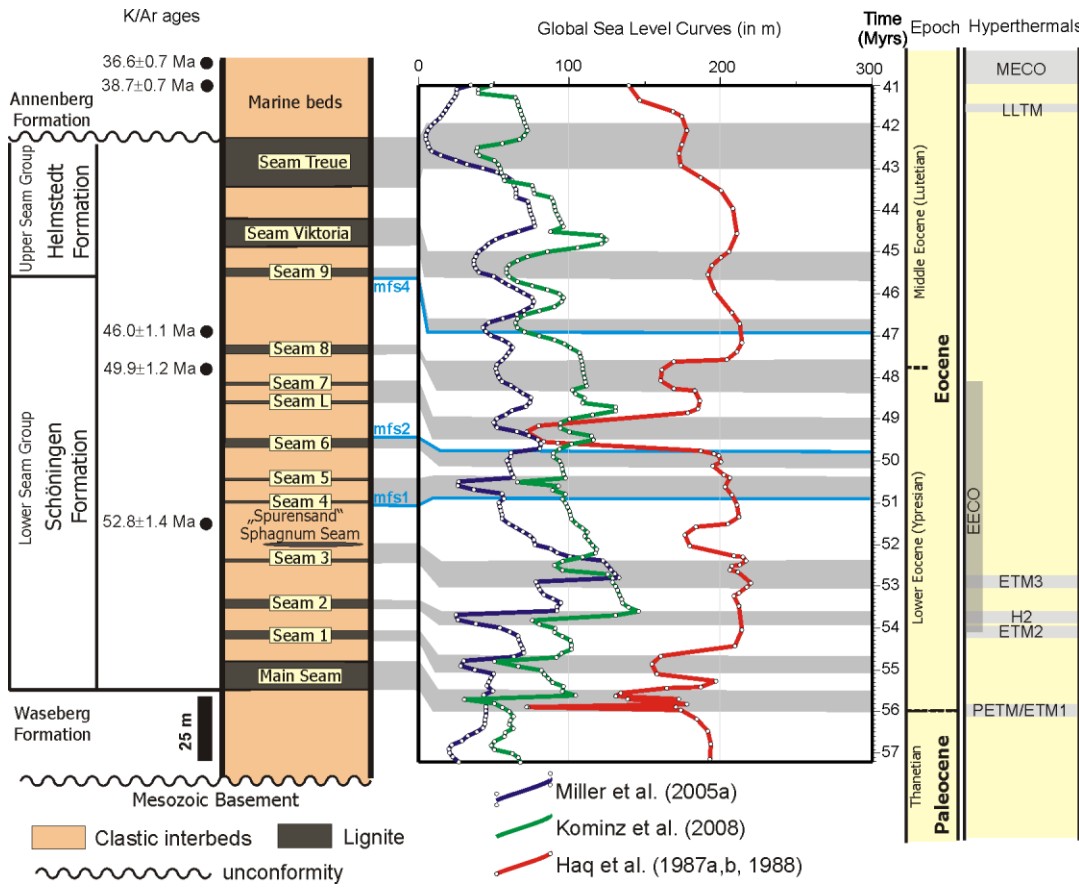

**Figure 3: Possible correlation of peat accumulation phases/lignite seams at Schöningen with lowstands of global sea level using the sea level curves of Haq et al. (1987a, b, 1988), Miller et al. (2005a), and Kominz et al. (2008). Ages for the Haq et al. (1987a, b, 1988)**

**curve are taken from Miller et al. (2005a). K/Ar ages have been taken from Ahrendt et al. (1995). Positions for maximum flooding surfaces (mfs 1, 2 and 4) were given by Osman et al. (2013). Mfs 3 cannot be indicated since it is related to the Emmerstedt Formation which is missing between the Schöningen and Helmstedt Formations at the Schöningen locality. Abbreviations of hyperthermals: PETM – Paleocene Eocene Thermal Maximum; ETM 1, 2, 3 – Eocene Thermal Maximum 1, 2, 3; H2 – H2 event; EECO – Early Eocene Climatic Optimum; LLTM – Late Lutetian Thermal Maximum; MECO – Middle Eocene Climatic**

**Optimum. Hyperthermal references are given in text.**





**Figure 4: Composite stratigraphic section of the lower part of the Schöningen Formation from the Main Seam up to the base of Interbed 7 with total organic carbon (%TOC) and δ¹³C values of bulk organic matter. Lines denote lithological changes between brackish interbeds and lignite seams. Six identified CIEs are indicated.**



**Figure 5: Details of total organic carbon (%TOC) and δ¹³C values of bulk organic matter presented in Fig. 4, illustrating variations of TOC and δ¹³C_TOC values at lithological boundaries between terrestrial lignite and estuarine clastic interbeds in the lower part of the Schöningen Formation. S = Seam, IB = Interbed.**



**Figure 6: Comparison of typical $\delta^{13}C$ records for the Paleocene-Eocene Thermal Maximum (PETM) in deep sea and continental/marginal marine records indicating that non-marine records commonly show a rebound to higher $\delta^{13}C$ values during the CIE body (black arrows). Data for the marine records: ODP Site 690, Maud Rise, South Atlantic (Nunes and Norris, 2006);**

**Mead Stream, New Zealand, South Pacific (Nicolo et al., 2010); ODP Site 1172, East Tasman Plateau, Tasman Sea (Sluijs et al., 2011). Data for continental and marginal marine records: Polecat Bench, Bighorn Basin, USA (Baczynski et al., 2013); Wasatch Formation, Piceance Creek Basin, Colorado, USA (Foreman et al., 2012), the black line corresponds to a 5-point running mean of bulk data; Mar 2X core, Venezulea (Jaramillo et al., 2010), the black line corresponds to a 5-point running mean of bulk data; Beigou section, Nanyang Basin, central China (Chen et al., 2014). Locations of the study sites are presented on a continental**

**reconstruction for the PETM (PALEOMAP project, Map 14; Scotese, 2014) Abbreviations for stable carbon isotope data: $\delta^{13}C_{DOM}$ - measurements of dispersed organic matter, $\delta^{13}C_{org}$ - measurements of organic matter; $\delta^{13}C_{DOC}$- measurements of dispersed organic carbon; $\delta^{13}C_{TOC}$ - measurements of bulk organic matter; $\delta^{13}C_{BC}$ - measurements of black carbon.**



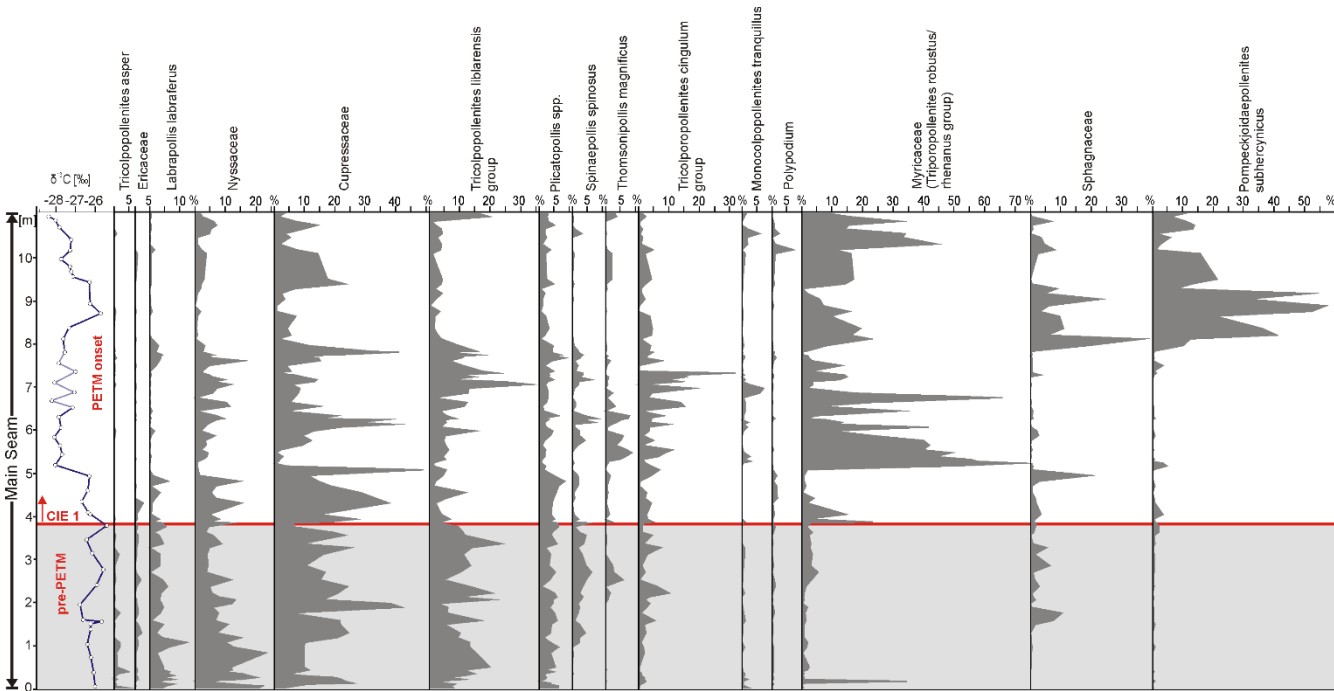

**Figure 7: Simplified pollen diagram of the Main Seam, showing frequency and palynological abundance changes between the pre-PETM and peak-PETM intervals. The carbon isotope data are shown for comparison. Due to the different thicknesses of the studied sections for isotopes and palynology, the isotope curve for the Main Seam has been tied to the top and base of the Seam. The red line indicates the beginning of the PETM-related CIE.**

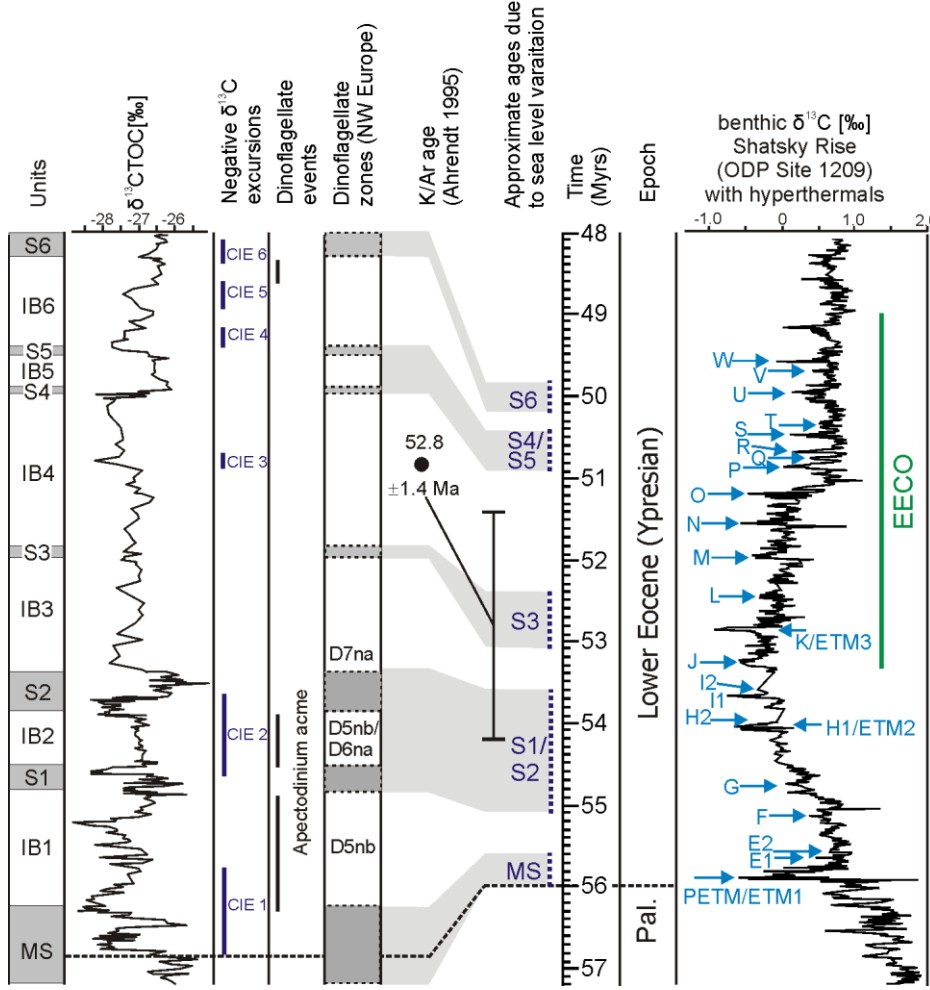

**Figure 8: Correlations between lignite and interbed units in the lower part of the Schöningen Formation, δ¹³C_TOC values (this work), dinoflagellate ages (this work), K/Ar age (Ahrendt et al. 1995), approximate ages of seams due to sea level fluctuations (this work) and δ¹³C isotopic excursion (CIE) ages (Westerhold et al. 2017, 2018b). Data for the benthic δ13C curve from the Shatsky rise (ODP site 1209, Northwest Pacific) as representing a typical deep marine isotope record for this time period are taken from Westerhold et al. (2018b). The comparison of carbon isotope records from Schöningen and the Northwest Pacific shows that CIE 1 is in the range of the PETM, CIE 2 in the range of the H1/ETM2 and H2 events and CIEs 3 to 6 in the range of small CIEs known from the marine record, which indicate high amplitude climate fluctuations during the EECO. Pal. = Paleocene, S = Seam, IB = Interbed.**