# Peer review of "Early Eocene carbon isotope excursions in a lignite bearing succession at the southern edge of the proto-North Sea (Schöningen, Germany)"

_Climate of the Past, 2021_

## Author Comment (AC1)

*This paper presents carbon isotope data of bulk organic matter (δ13CTOC), organic carbon content, and palynomorphs from an Early Eocene study site in Germany. The authors provide a new age model for the section and despite the variability in carbon isotope values, try to make educated interpretations to outline up to six potential negative carbon isotope excursions (CIEs). A full assessment of the intricacies of the lithological interpretations and stratigraphic framework is outside my area of expertise. That said, the paper appears to provide a lengthy description of observations.*

**We thank the reviewer for the careful and constructive comments and suggestions to improve the paper. In the following each of the comments are specifically addressed.**

**Reviewer comment 1:** *One particular challenge is that there is a lot of variability in the $δ^{13}C_{TOC}$ values, which is a problem with relying on bulk organic matter to define carbon isotope excursion intervals. The authors do provide some context relative to other sections, especially with regards to the variability in the isotope values. I cannot help but wonder if others looking at the carbon isotope values would draw the same conclusions because of the variability.*

**Our response:** In conjunction with our age model, which should be very robust, especially due to the occurrence of stratigraphically relevant dinocysts (see also our response to comment 2), we are able to calibrate the isotope curve quite precisely, especially in the lower part. Therefore, the interpretation of our CIE 1 as PETM-CIE is conclusive, especially in comparison to similar curves of other sections at the southern edge of the Proto-North Sea also relying on bulk organic matter. The interpretation of the other post-PETM CIEs has been made more cautiously but is also supported by our age model.

**Reviewer comment 2:** *It does seem like this paper is a helpful addition to constraining an additional study site of Early Eocene CIEs. This would be constructive for future work analyzing this area and for comparison to study sites in other regions to understand climate in the past. I think it's important to better note the level uncertainty in the results and age correlations to avoid oversimplification though.*

**Our response:** We will add a new reference (Iakovleva et al. 2021). The authors notice that the dinocyst assemblages from the PETM interval in the Paris and Dieppe-Hampshire basins at the southern edge of the Proto-North Sea contain a significant number of atypical, longer specimens of *Apectodinium parvum*, which could represent an ecological onshore substitute for *Axiodinium augustum,* the open marine marker species of the PETM. Since *A. parvum* is together with *A. homomorphum* the dominant cyst among the dinoflagellate assemblage at the base of interbed 1, this is further evidence that the CIE, which starts in the Main Seam, represents the PETM. We will therefore update the dinocyst photo plate in the supplementary material and the discussion of our age model in the manuscript accordingly.

**Reviewer comment 3:** *Has any attempt been made to do compound-specific isotopes to help define the carbon isotope excursions? I think it would be worth mentioning this in the paper. For example, Ln 423 refers to work from the Bighorn Basin, where more recent work by Baczynski and others have provided evidence for how bulk organic matter carbon isotopes values can be distorted indicators of CIEs (as a function of a combination of organic matter sources, reworking of organic matter, and degradation). See for example, Baczynski et al. 2016 https://doi.org/10.1130/B31389.1*

**Our response:** At the moment, we can only present the analysis of bulk organic matter carbon isotopes, which is also the standard method for other records at the southern edge of the Proto-North Sea. We are fully aware of the problem such as reworking of organic matter, differential degrading of organic matter, and the combination of different organic matter sources as already discussed in Methner et al. (2019) (see also our response to comment 4). Therefore, we only interpreted general trends of the curve. We are aware of the mismatch between isotopes values of bulk organic matter and compound-specific isotopes, but this does not affect our interpretation, since the CIEs that we recognised should actually be even more pronounced by using compound-specific isotopes. In that context we will now refer to Baczynski et al. (2016) and add the reference.

**Reviewer comment 4:** *Ln 99. The frequent changes between terrestrial and marine conditions and thus changes in TOC input type can be a challenge for sourcing. Mixed inputs would affect the $\delta^{13}C$ of the bulk organic matter.*

**Our response:** We are fully aware of the problem of mixing of different carbon sources. Therefore, we have already written a paragraph (4.3 - Effects of changes in lithology and mixing of carbon sources on $\delta^{13}C_{TOC}$ values) in which we discuss this problem.

**Reviewer comment 5:** *Ln 165. The figure number of the diagram needs to be specified.*

**Our response:** We didn't specify the figure number in Ln. 165, since the pollen diagram should appear as Fig. 7 at the end of the text. Otherwise, because of consecutive numbering of all figures, the pollen diagram would have to be inserted as Fig. 2 at the wrong place in the text. Therefore, we would suggest omitting the reference to the figure number at this point.

**Reviewer comment 6:** *Ln 276-281. How does the variability compare to other well-constrained CIEs and other sections with similar lithology changes as this section? In Ln 279, "comparatively high standard deviation" – what are the authors comparing to? What is causing the high variability, the "great range of $\delta^{13}C_{TOC}$"?*

**Our response:** At this point in the text we only present the results of our isotope analyzes and describe the course of the $\delta^{13}C$ curve. The comparison with other well-constrained CIEs in other locations with similar lithology, e.g., the Cobham Lignite (UK) and the Vasterival section (France), is discussed in detail in paragraph 4.4.1.

We will change the sentence in line 279 in "…with the comparatively high standard deviation (SD), compared to other sections of the isotope record (see below),…". Additionally, we will add the reference to Table 2 to present the "great range" of $\delta^{13}C_{TOC}$ values in this part of the record.

Furthermore, we will extend the discussion in paragraph 4.4.1 and add a new figure providing a comparison to other PETM-CIEs from lignite-bearing deposits at the southern margin of the paleo-North Sea such as Cobham, Vasterival and Northern Belgium. In addition, a continental isotope curve from the Pyrenees (Claret section) is also shown for comparison. The $\delta^{13}C$ values from this section, which does not contain any lignites, are clearly higher than those from the wetlands of the North Sea. Only the $\delta^{13}C$ values from the Doel borehole in Belgium are comparable to those from the Pyrenees. However, the section from Belgium includes only few terrestrial sediments, is mainly lagoonal to marine influenced and therefore not fully comparable to the sections from Cobham, Vasterival and Schöningen. Especially in comparison with Vasterival, the variability of the $\delta^{13}C$ values in the pre-, peak-, and post-CIE intervals is almost identical.

We pointed to the similarities between the lignite deposits at Cobham, Vasterival and Schöningen (CIE 2 in our manuscript) already in Methner et al. (2019):

1. Absolute $\delta^{13}C_{TOC}$ values and the range in $\delta^{13}C_{TOC}$ values (-3:2 ‰) are very similar.
2. All three records attain similar minimum $\delta^{13}C_{TOC}$ values during the CIE (-27.5‰ to -28.8‰), averaging $28.05\pm0.5$‰.
3. At the onset of the CIE the magnitude of changes in $\delta^{13}C_{TOC}$ (calculated as the difference between the last pre-CIE value and the first CIE value) ranges only between -1.4‰ and -1.8‰.
4. Magnitudes of the CIE in bulk organic matter calculated as the difference between the mean pre-CIE and the mean CIE values range from -0.9‰ to -1.6‰. CIEs calculated as the difference between the mean pre- CIE values and the most negative value during the CIE yield magnitudes of -1.1‰ to -2.6‰.

These characteristics now also apply to our CIE 1, which we correlate with the PETM. In summary, the very similar data from the lignite records show that the wetlands at the southern edge of the paleo-North Sea had a uniform behavior during latest Paleocene-early Eocene thermal events. The CIE magnitudes are damped compared to purely continental terrestrial archives such as the Claret section in the Pyrenees but yield a very consistent and robust signal. In Methner et al. (2019) we discuss different possibilities for the dampened magnitude of the CIE in the lignite records such as mixing and

dilution of the input signal, occurrence of local signal perturbation (e.g., due to vegetation changes), or the differential degradation and/or preservation of organic matter during the climatic perturbation.

[Figure]

**New figure:** Comparison of midlatitudinal $\delta^{13}C_{TOC}$ records surrounding the paleo-North Sea: Cobham, UK (Collinson et al., 2003), Vasterival, France (Storme et al., 2012), Schöningen, Germany (this study), Doel borehole, Northern Belgium (Steurbaut et al. 2003). Additionally, a terrestrial $\delta^{13}C_{TOC}$ record from the Pyrenees (Claret section, Domingo et al. 2009) is given that highlights the isotopic differences to the lignite records. Note the different stratigraphic thicknesses due to different sediment accumulation and preservation conditions in the individual depositional environments.

**Reviewer comment 7:** *Ln 280. "significant decrease". On what statistical basis was this determined to be significant?*
**Our response:** This statement is not statistically based but relative. Therefore, we will change "significant" in "strong".

**Reviewer comment 8:** *Ln 333. "values increase significantly" As above, are there statistics to attest to the significance?*
**Our response:** We will delete "significantly" here.

**Reviewer comment 9:** *Ln 342 "a weak CIE". Weak is a qualitative term, information on the magnitude of the excursion/time or depth interval/number of samples could help clarify the identification of this CIE.*
**Our response:** We will add "of 0.7 ‰" ("a weak CIE of 0.7 ‰").

**Reviewer comment 10:** *Ln 389. I think the authors meant 1,000 years (1 kyr), not 1.000 years.*
**Our response:** Thanks for pointing that out. We will write "1 kyr".

**Reviewer comment 11:** *Ln 422. Can you provide and clarify the evidence that rules out reworking of organic matter? This is again stated in the conclusions, Ln 489, reworking is almost excluded within our seam. As far as I can tell, the logic may be that the rebound structure is consistent with the pollen and spore record…but that doesn't mean reworking couldn't also be a factor?*
**Our response:** We will extend the discussion on reworking of organic matter and the rebound structure in our isotope record of the Main Seam and write in paragraph 4.1.1:
"Since within-seam carbon sources are likely to be purely terrestrial, the fluctuations of $\delta^{13}C_{TOC}$ values in the CIE of the Main Seam including a clear rebound to higher values are striking. The PETM reflects injections of large amounts of carbon into the atmosphere-ocean system, but it is uncertain, whether single or multiple episodes of carbon release caused this event (Zhang et al. 2017). Generally, the PETM-related CIE is triangular with a single large decrease in $\delta^{13}C$ values, followed by an exponential recovery as seen in several marine isotope curves (Fig. 6). However, in some records, $\delta^{13}C$ values decrease in several steps indicating complex processes of carbon injection into the atmosphere (Zhang et al. 2017). These steps can be interpreted as a result of multiple phases of carbon release. Bowen et al. (2015) could show in a high-resolution carbon isotope record of terrestrial deposits in the Bighorn Basin (USA) that the beginning of the PETM consists of at least two discrete intervals of decreasing carbon isotope values. The first event had a duration between 2 and 5.5 kyr and the onsets of the two carbon release events lasted not more than 1.5 kyr (Bowen et al. 2015). In our record, the first $\delta^{13}C$ decline was also within less than 1.5 kyr and the first excursion lasted around 4.5 kyr when assuming that one meter of lignite represent about 1 kyr (see above*). Therefore, the data from the Main Seam are consistent with the excursions reported from the Bighorn Basin.
However, the steps in a carbon isotope curve may also be related to sediment reworking or mixing of different carbon sources (e.g., Bralower 2002). Mixing and reworking of organic matter cannot be excluded for the interbeds of the Schöningen succession, since multiple flooding and thus reworking events may have occurred (Methner et al. 2019). However, within the autochthonous lignite seams no mixing of the organic material occurred (Methner et al. 2019). The organic matter of the lignite results in situ from an ombrotrophic (rain-fed) peat mire consisting mostly of mosses, ferns and an associated hardwood mire forest (Riegel et al. 2012, Inglis et al. 2015, Lenz et al. 2021). Therefore, the isotope curve in the Main Seam should indeed reflect a stepwise injection of carbon into the atmosphere."

*According to an average compaction rate of 3:1 (Widera 2015) and an average rate of deposition for tropical peat of 3 mm/year (Stach et al. 1982) as written at the beginning of the paragraph.

**Reviewer comment 12:** *Table 1. What do the parentheses indicate vs. non-parentheses for the references? Also check for consistent spacing in the table between words.*
**Our response:** In botanical nomenclature, in cases where a species is no longer in its original generic placement (i.e., a new combination of genus and specific epithet), both the authority for the original genus placement and that for the new combination are given (the former in parentheses). The mistakes regarding spacing between words will be fixed.

**Reviewer comment 13:** *Table 2. It would be useful to specify what stratigraphic positions (meter levels) were used for each of the identified excursions.*
**Our response:** We will add a new column "Stratigraphic position (in m above base of the section)" in the table.

**Reviewer comment 14:** *Figure 1. It looks like there is an extra space in the "Seam 7" label.*
**Our response:** Thanks for pointing that out. We will fix the bug in the figure.

**Reviewer comment 15:** *Figure 3. Is there any meaning between the different number of dots in the lines in the legend for global sea level curves?*
**Our response:** Thanks for pointing that out. We will fix the bug in the figure.

**Reviewer comment 16:** *Figure 6. I realize the authors may be citing the units used in the other publications, but it's unclear to the reader whether there is any meaningful difference between the types of organic matter. For example, $\delta^{13}C_{org}$ "organic matter" versus $\delta^{13}C_{TOC}$ "bulk organic matter".*
**Our response:** This is correct. We will change the units in all terrestrial records in "$\delta^{13}C$", since all measurements are based on bulk organic matter. Only the isotope measurements of the Chinese record are based on measurements of black carbon. Therefore, we will add in the figure caption "…; in contrast to standard measurements of $\delta^{13}C$ (bulk organic matter) not only carbonates were removed from the samples prior to $\delta^{13}C_{BC}$ measurements but also silicates by using hydrofluoric acid…".
Additionally, a wrong reference was given for the Polecat bench record. We will change Baczynski et al. (2013) in Bataille et al. (2013)

**Reviewer comment 17:** *Figure 7. Clarify is the carbon isotope data here from bulk organic matter?*
**Our response:** We will change the sentence in the figure caption "The carbon isotope data are shown for comparison." in "The $\delta^{13}C$ values of bulk organic matter (see Fig. 4 for the complete data set) are shown for comparison."

**Reviewer comment 18:** *Figure 8. TOC for the carbon isotope record should be subscript.*
**Our response:** We will change it accordingly.

References:
Baczynski, A. A., McInerney, F. A., Wing, S. L., Kraus, M. J., Morse, P. E., Bloch, J. I., Chung, A. H., and Freeman, K. H.: Distortion of carbon isotope excursion in bulk soil organic matter during the Paleocene-Eocene thermal maximum, GSA Bulletin; 128, 1352–1366, https://doi.org/10.1130/B31389.1, 2016.
Bataille, C. P., Mastalerz, M., Tipple, B. J., and Bowen, G. J.: Influence of provenance and preservation on the carbon isotope variations of dispersed organic matter in ancient floodplain sediments, Geochem. Geophys. Geosyst., 14, 4874–4891, https://doi.org/10.1002/ggge.20294, 2013.
Bowen, G., Maibauer, B., Kraus, M., Röhl, U., Westerhold, T., Steimke, A., Gingerich, P. D., Wing, S. L., and Clyde, W. C.: Two massive, rapid releases of carbon during the onset of the Palaeocene–Eocene thermal maximum, Nature Geoscience, 8, 44–47, https://doi.org/10.1038/ngeo2316, 2015.
Bralower, T.J.: Evidence of surface water oligotrophy during the Paleocene-Eocene thermal maximum: Nannofossil assemblage data from Ocean Drilling Program Site 690, Maud Rise. Weddell Sea. Paleoceanography. http://dx.doi.org/10.1029/2001pa000662, 2002
Collinson, M. E., Hooker, J. J., and Gröcke, D. R.: Cobham Lignite Bed and penecontemporaneous macrofloras of southern England: A record of vegetation and fire across the Paleocene-Eocene Thermal Maximum, in: Causes and consequences of globally warm climates in the early Paleogene, edited by: Wing, S. L., Gingerich, P. D., Schmitz, B., and Thomas, E., GSA Special Paper, 369, 333–349, https://doi.org/10.1130/0-8137-2369-8.333, 2003.
Domingo, L., López-Martínez, N., Leng, M. J., and Grimes, S. T.: The Paleocene–Eocene Thermal Maximum record in the organic matter of the Claret and Tendruy continental sections (South-central Pyrenees, Lleida, Spain), Earth and Planetary Science Letters 281, 226–237, https://doi.org/10.1016/j.epsl.2009.02.025, 2009

Iakovleva, A. I., Quesnel, F., and Dupuis, C.: New insights on the Late Paleocene - Early Eocene dinoflagellate cyst zonation for the Paris and Dieppe basins, Earth Sciences Bulletin, 192, 44, https://doi.org/10.1051/bsgf/2021035, 2021.

Inglis, G. N., Collinson, M. E., Riegel,W.,Wilde, V., Robson, B. E., Lenz, O. K., and Pancost, R. D.: Ecological and biogeochemical change in an early Paleogene peat-forming environment: Linking biomarkers and palynology, Palaeogeogr., Palaeoecl., 438, 245– 255, 2015Lenz, O. K, Riegel, W., and Wilde V.: Greenhouse conditions in lower Eocene coastal wetlands? - Lessons from Schöningen, Northern Germany, PLoS ONE, 16, e0232861, https://doi.org/10.1371/journal.pone.0232861, 2021.

Methner, K., Lenz, O. K., Riegel, W., Wilde, V., and Mulch, A.: Paleoenvironmental response of midlatitudinal wetlands to Paleocene–early Eocene climate change (Schöningen lignite deposits, Germany), Clim. Past, 15, 1741–1755, https://doi.org/10.5194/cp-15-1741-2019, 2019.

Riegel, W., Wilde, V., and Lenz, O. K.: The early Eocene of Schöningen (N-Germany) – an interim report, Austrian J. Earth Sci., 105, 88–109, 2012.

Stach, E., Mackowsky, M. T., Teichmüller, M., Taylor, G. H., Chandra, D., and Teichmüller, R.: Stach's Textbook of Coal Petrology, Gebrüder Borntraeger, Berlin, Germany, 1982

Steurbaut, E., Magioncalda, R., Dupuis, C., Van Simaeys, S., Roche, E., and Roche, M.: Palynology, paleoenvironments, and organic carbon isotope evolution in lagoonal Paleocene-Eocene boundary settings in North Belgium, in: Causes and consequences of globally warm climates in the early Paleogene, edited by: Wing, S. L., Gingerich, P. D., Schmitz, B., and Thomas, E., GSA Special Paper, 369, 291–317, https://doi.org/10.1130/0-8137-2369-8.291, 2003.

Storme, J. Y., Dupuis, C., Schnyder, J., Quesnel, F., Garel, S., Iakovleva, A. I., Iacumin, P., Di Matteo, A., Sebilo, M., and Yans, J.: Cycles of humid-dry climate conditions around the P/E boundary: new stable isotope data from terrestrial organic matter in Vasterival section (NW France), Terra Nova, 24, 114–122, https://doi.org/10.1111/j.1365-3121.2011.01044.x, 2012.

Widera, M.: Compaction of lignite: a review of methods and results. Acta Geol Pol, 65, 367–378, https://doi.org/10.1515/agp-2015-0016, 2015.

Zhang, Q., Wendler, I., Xu, X., Willems, H., and Ding, L.: Structure and magnitude of the carbon isotope excursion during the Paleocene-Eocene thermal maximum, Gondwana Research 46, 114–123, http://dx.doi.org/10.1016/j.gr.2017.02.016, 2017.

---

## Author Comment (AC3)

*The study by Lenz and colleagues provides a high-resolution terrestrial organic carbon isotope curve covering the Late Paleocene – Early Eocene. The new d$^{13}$Corg curve is embedded into a detailed stratigraphic framework based on new dinoflagellate biostratigraphy coupled with sequence-stratigraphic considerations. The new age constraints enable comparison (and tentative correlation) of pronounced carbon isotope anomalies (CIE 1 to 6) observed in the lignite-bearing record with marine carbon isotope trends and associated hyperthermals including the PETM, ETM2 (questionable) and EECO. The core of the study is a high-resolution δ$^{13}$C$_{org}$ curve based on >320 measurements, part of which (~120 data points) has already been published by Methner et al. (2019) in Climate of the Past.*

*The manuscript represents a well written scientific study of very good quality, presenting new and important findings. The data-set is well presented in a number of high-quality figures and diagrams. Stratigraphically well constrained land-sea correlations during times of exceptional global warmth are of paramount importance to better understand the coupled response of the marine and continental biosphere during such hyperthermal events. In this respect, the study is clearly well suited for publication in Climate of the Past. However, some aspects remain critical and need revision, as outlined below.*

**We thank the reviewer for the careful and constructive comments and suggestions to improve the paper. In the following each of the comments are specifically addressed.**

Major points

**Reviewer comment 1:** *My main criticism is the absence of an in-depth discussion on the controls of the carbon isotopic composition of the mire-derived OM, which is interpreted here to record a global atmospheric carbon isotope signal. In my view, several aspects should be considered in order to better distinguish between potential global and local carbon isotope signatures. Firstly, the overall composition of the predominantly land plant-derived OM is rather negative (-25.5 to -28.5 ‰) which deserves some discussion. Gröcke (2002) gives an average of -23 to -27 for C3 land plant-derived OM. Comparison with time-equivalent plant derived carbon isotope values would help to get an idea of the background level expected during the Late Paleocene – Early Eocene. In addition, data from other nearshore mire deposits would provide a range for the carbon isotope composition variability in such environments.*

**Our response:** We will extend the discussion in paragraph 4.4.1 and will add a new figure providing a comparison to other PETM-CIEs from lignite-bearing deposits at the southern margin of the paleo-North Sea such as Cobham, Vasterival and Northern Belgium. In addition, a continental isotope curve from the Pyrenees (Claret section) is also shown for comparison. The δ$^{13}$C values from this section, which does not contain any lignites, are clearly higher than those from the wetlands of the North Sea. Only the δ$^{13}$C values from the Doel borehole in Belgium are comparable to those from the Pyrenees. However, the section from Belgium includes only few terrestrial sediments, is mainly lagoonal to marine influenced and therefore not fully comparable to the sections from Cobham, Vasterival and Schöningen. Especially in comparison with Vasterival, the variability of the δ$^{13}$C values in the pre-, peak and post-CIE intervals is almost identical.

We pointed to the similarities between the lignite deposits at Cobham, Vasterival and Schöningen (CIE 2 in our manuscript) already in Methner et al. (2019):

1. Absolute δ$^{13}$C$_{TOC}$ values and the range in δ$^{13}$C$_{TOC}$ values (-3:2 ‰) are very similar.
2. All three records attain similar minimum δ$^{13}$C$_{TOC}$ values during the CIE (-27.5‰ to -28.8‰), averaging 28.05$_{+}$0.5‰.
3. At the onset of the CIE the magnitude of changes in δ$^{13}$C$_{TOC}$ (calculated as the difference between the last pre-CIE value and the first CIE value) ranges only between -1.4‰ and -1.8‰.
4. Magnitudes of the CIE in bulk organic matter calculated as the difference between the mean pre-CIE and the mean CIE values range from -0.9‰ to -1.6‰. CIEs calculated as the difference between the mean pre- CIE values and the most negative value during the CIE yield magnitudes of -1.1‰ to -2.6‰.

These characteristics now also apply to CIE 1, which we correlate to the PETM in the present study. In summary, the very similar data from the lignite records show that the wetlands at the southern edge of the paleo-North Sea had a uniform behavior during latest Paleocene-early Eocene thermal events. The CIE magnitudes are damped compared to purely continental terrestrial archives such as the Claret section in the Pyrenees but yield a very consistent and robust signal. In Methner et al. (2019) we discussed different possibilities for the dampened magnitude of the CIE in the lignite records such as mixing and dilution of the input signal, occurrence of local signal perturbation (e.g., due to vegetation changes), or the differential degradation and/or reservation of organic matter during the climatic perturbation.

[Figure]

**New figure:** Comparison of mid-latitudinal $\delta^{13}C_{TOC}$ records surrounding the paleo-North Sea: Cobham, UK (Collinson et al., 2003), Vasterival, France (Storme et al., 2012), Schöningen, Germany (this study), Doel borehole, Northern Belgium (Steurbaut et al. 2003). A terrestrial $\delta^{13}C_{TOC}$ record from the Pyrenees (Claret section, Domingo et al. 2009) is given in addition that highlights the isotopic differences to the lignite records. Note the different stratigraphic thicknesses due to different sediment accumulation and preservation conditions in the individual depositional environments.

**Reviewer comment 2:** *In chapter 4.3, the authors refer to the general difference in the $\delta^{13}C$ composition of land- and marine-derived OM. In consequence, mixing of marine and terrestrial organic*

*carbon is used to explain certain variations in the Schöningen stratigraphic record (line 321 ff.). However, when comparing the δ¹³C signature of isolated marine particles (dinoflagellate cysts) analyzed from before and throughout the PETM (Sluijs et al. 2017, Geology), it becomes clear that the dinoflagellate-derived δ¹³C composition covers a similar range compared to the OM from the Schöningen record. During the Early Paleogene, the δ¹³Corg composition of marine OM is not depleted compared to the values obtained from Schöningen, which hampers any source assignment of the OM based on the δ¹³C signature. To better distinguish between different sources, RockEval pyrolysis and/or palynofacies data would certainly help.*

**Our response:** It is correct that the $\delta^{13}C$ signature of isolated dinoflagellate cysts as reported by Sluijs et al. (2018) covers a similar range compared to the Schöningen record, but our bulk organic matter may include other marine organic matter. For example, Sluijs et al. (2018) present data of marine amorphous organic matter that is very depleted in $^{13}C$. We will therefore extend the respective paragraph and write:

"Since marine organic matter is depleted in $^{13}C$ (Sluijs and Dickens, 2012) mixing of $^{13}C$ from marine and terrestrial sources will influence $\delta^{13}C_{TOC}$ values. By comparing the $\delta^{13}C$ signature of isolated dinoflagellate cysts analyzed from before and throughout the PETM (Sluijs et al. 2018) with $\delta^{13}C_{TOC}$ values from the Schöningen Formation, it becomes clear that the $\delta^{13}C$ composition of dinocysts covers a similar range. However, the marine bulk organic matter that has been analyzed here, also contains organic compounds that may have been strongly depleted in $^{13}C$ such as amorphous organic matter shifting the $\delta^{13}C_{TOC}$ record to lower values (see Sluijs et al. 2018)."

RockEval pyrolysis and/or palynofacies data are indeed possibilities to distinguish between different sources of organic matter but beyond the scope of the present paper. However, a detailed palynofacies study is ongoing and preliminary results already exist for minor parts of the record (e.g., Seam 1, Lenz et al. 2021). However, the data are far from having been completed yet for the lower part of the Schöningen Formation.

**Reviewer comment 3:** *At the beginning of chapter 4.3 (line 309-310), the authors state that when comparing the new CIEs with known CIEs, a differentiation between shifts at lithological boundaries and shifts occurring within the same lithology is necessary. This statement seems to indicate that CIEs associated with lithological boundaries are more prone to be caused by local changes (e.g. OM composition) compared to within-facies shifts, which are – in consequence - interpreted as super-regional (global) phenomena. The authors do not refer to potential processes, which may cause the high-amplitude shifts in δ¹³Corg within individual coal seams. Their data shows that pronounced changes do occur within many of the studied coal seams, which may be controlled by various environmental processes and not necessarily reflect changes in the global δ¹³C signature. Mires do evolve with time and mire-producing plant successions change with mire growth, which is expected to cause stratigraphic changes in the bulk δ¹³C signature of the peat/lignite. Interestingly, the detailed pollen record from the main seam (Fig. 7) does show pronounced changes in the pollen assemblage, which does occur time-equivalent to a major shift in the δ¹³C record (e.g. pronounced increase of Myricaceae associated with a strong negative CIE). This negative CIE is considered to represent the onset of the PETM negative CIE and therefore, the co-occurrence of negative CIE and vegetation shift needs to be explained in more detail. In lines 411 ff. the authors refer to this coincidence as intra-PETM fluctuations, but a more in-depth discussion is lacking. The authors need to explain, on which basis local environmental drivers (incl. vegetation and/or humidity changes) can be excluded to explain the onset of the negative CIE 1. In general, potential environmental processes affecting the δ¹³C composition of the OM need to be considered in more depth and need to be included in the interpretation of the stratigraphic trend obtained from Schöningen.*

**Our response:**

We absolutely agree with the reviewer's note that mires evolve with time and mire-producing plant communities change with mire growth. The peat forming plant communities of the lower Schöningen Formation are a perfect example for this. So far, high-resolution palynological studies are available for Seam 1 (Lenz et al. 2021) and Seam 2 (Methner et al. 2019) and expanded here to the Main Seam. All three seams are characterized by highly variable pollen assemblages.

We also agree with the reviewer that it is of crucial importance to prove whether vegetation changes and related $\delta^{13}C$ variations in the record of the Schöningen Formation are caused by changes of the Paleogene climate (global signal) or induced by other probably regional factors. However, a climatic influence and the corresponding response of the ecosystems cannot simply be revealed from the microflora as documented in our studied successions. Multiple alternations of lignites with marginal marine and fluvial interbeds in the section indicate significant facies changes which should have been coupled with changes in vegetation. When studying climate changes and perturbations in the palynomorph record it is therefore necessary to distinguish between changes in vegetation that were controlled by climate or other factors such as natural succession due to peat aggradation.

The isotope analyses from the lower part of the Schöningen Formation revealed an excursion ranging from the top of Seam 1 to the middle of Seam 2 (Methner et al. 2019, CIE 2 in the present manuscript). However, carbon isotope values do not indicate a CIE for Seam 1 except for the uppermost samples. Therefore, Seam 1 has been deposited almost completely during a period without strong climate perturbations and changes in the vegetation were only controlled by factors other than climate. Therefore, Seam 1 was selected in a recent study (Lenz et al. 2021) to study the composition and variability of the regional flora beyond warming events. In spite of low variability in isotope values considerable changes of the vegetation have been observed here:

Seam 1 (Lenz et al. 2021) shows a distinctive threefold succession of the vegetation during formation of the seam: an initial, a transitional and a terminal stage. The seam is sandwiched between marginal marine Interbeds 1 and 2. Therefore, in total four different types of peat depositional environments and vegetation have been distinguished for Seam 1 representing the natural succession of plants in the early Eocene coastal wetland: (1) a coastal vegetation, (2) an initial mire, (3) a transitional mire and (4) a terminal mire.

The same natural succession is observed for Seam 2 (Methner et al. 2019) and now for the Main Seam (this study), revealing that the natural succession as seen in Seam 1 is typical for the evolution of the peat forming mire forest at the edge of the southern proto-North Sea during this time and reflects vegetation responses to changes in environment and facies that took place during an early Paleogene regression/transgression cycle including the formation of peat. Noteworthy is a pronounced change in the palynomorph assemblage composition that occurs in the upper part of the three seams, which represents a terminal mire at the end of peat accumulation. This is mainly due to the rise to dominance of *Sphagnum*-type spores. Furthermore, a great increase in pollen of the Myricaceae can be recognized. Together with *Sphagnum* they clearly signal that peatbeds in the terminal mire phase were decoupled from groundwater and their hydrology was increasingly controlled by precipitation. This shows that regional factors play a major role for the composition of the vegetation.

As the reviewer noted, in the Main Seam (this study), the occurrences of myricaceous pollen and *Sphagnum* spores clearly correlate with the $\delta^{13}C$ values. The strong increase of myricaceous pollen with a simultaneous decrease of *Sphagnum* spores is associated with a strong decrease of $\delta^{13}C$ values, while at the top of the seam the increase of $\delta^{13}C$ values is associated with a strong increase of *Sphagnum* spores and *Pompeckjoidaepollenites subhercynicus*. If the $\delta^{13}C$ signal is a purely regional signal that reflects environmental processes in the mire forest, the isotope values in Seams 1 and 2 should behave similarly. In Seam 1, however, the $\delta^{13}C$ signal remained constant during the increase of *Sphagnum* spores and *P. subhercynicus* during the terminal phase, and the strong decrease in the $\delta^{13}C$ values at the top of the seam was only evident after the increase of myricaceous pollen started (see Methner et al. 2019). Also in Seam 2 the values of myricaceous pollen and *P. subhercynicus* are not correlated with the isotope values. Only the increase in *Sphagnum* spores is associated with an increase in $\delta^{13}C$ values, which according to Methner et al. (2019) coincides with the end of a CIE. Therefore, main changes in the peat-forming vegetation only correlate to the isotope signal in the Main Seam, but not in Seam 1 and Seam 2. Local processes therefore have no strong influence on the isotope signal in Seam 1 and 2. Although the vegetational changes are fundamentally controlled by regional environmental changes, the pattern of $\delta^{13}C$ values in the Main Seam, which differs significantly from that in the other two seams, indicates that global processes such as climate change during the PETM also have played an additional role during deposition of the Main Seam. Accordingly, the extreme increase of myricaceous pollen in the Main Seam, which is much stronger compared to

their increase in Seams 1 and 2, may indicate that climatic changes have enhanced environmental changes within the peat-forming vegetation.

A detailed discussion of potential environmental processes affecting the $\delta^{13}C$ composition has already been given by Methner et al. (2019). However, we will extend the discussion by the above described comparison of palynological and carbon isotope changes within the Main Seam and Seams 1 and 2. Furthermore, we will change the last paragraph of chapter 4.4.1. and write:

"A comparable shift from a mixed angiosperm-conifer flora in the Paleocene to an angiosperm flora at the onset of the PETM-related CIE was already reported for the Bighorn Basin in Wyoming (Smith et al., 2007, Diefendorf et al., 2010). Diefendorf et al. (2010) thereby suggested that a 4.6 ‰ decline in $\delta^{13}C$ values of atmospheric $CO_2$ at the onset of the PETM can be attributed to the different carbon isotope budget of gymnosperm and angiosperm leaves. Conifers and angiosperms reacted differently in isotopic fractionation during the PETM-CIE resulting in different isotopic excursions of ca. 3‰ for conifers and 6‰ for angiosperms (Schouten et al. 2007). Therefore, the strong decline of $\delta^{13}C_{TOC}$ values at 5.20 m in the Main Seam may also be attributed to the strong increase of Myricaceae in the wetland vegetation. However, at around 8.20 m a further increase in angiosperm pollen (*P. subhercynicus*) and the simultaneous decrease in pollen from the Cupressaceae is not associated with a decrease in $\delta^{13}C_{TOC}$ values but with a shift to less negative values. Therefore, a local change in the vegetation that altered the carbon isotope signal is unlikely to account solely for the CIE in the Main Seam."

Additionally, we will add a respective figure (see below) in the supplementary material that summarises isotope data and palynological data between the base of the Main Seam and the top of Seam 2.

[Figure]

**New supplementary figure:** Simplified pollen diagram of the base of the Main Seam up to the top of Seam 2, showing palynological abundance changes of dominant taxa. Due to the different thicknesses of interbed sections that were studied for palynology and carbon isotopes, $\delta^{13}C_{TOC}$ values of the interbeds have been tied to the top of the Main Seam and the base of Seam 1 as well as the top of Seam 1 and the base of Seam 2. Data for the part between the top of the Main Seam and the top of Seam 2 are taken from Methner et al. (2019).

**Reviewer comment 4:** *Chapter 4.4.1 deals with the CIE associated with the PETM. The authors provide compelling evidence for a correlation of the Schöningen CIE1 with the globally recognized PETM CIE. What is less clear is the basis for the base and top boundaries show in Fig. 6, which constrain the PETM CIE. Why is the basal boundary placed at the data point showing the least*

*negative value in this part of the curve (and not slightly higher at the data point just before the negative shift)? This has implications for the total amplitude of the anomaly (see table 2). Similarly, the positioning of the upper limit (red line in Fig. 6) of the CIE is hard to follow given that another interval with comparatively negative values is following above. A more plausible termination of the CIE would be at the transition towards less negative values (~23 m height). Are there any stratigraphic constraints for the positioning of those boundaries?*

**Our response:** The PETM reflects injections of large amounts of carbon into the atmosphere-ocean system, but, whether single or multiple episodes of carbon release caused this event is uncertain. We think that we have a robust example, which shows that the beginning of the PETM consists of more than one discrete interval of decreasing carbon isotope values as shown by Bowen et al. (2015) and Tremblin et al. (2022). Zhang et al. (2017) present an example that shows that the PETM is associated with a stepped CIE curve indicating complex processes of carbon injection into the atmosphere. As discussed in Zhang et al. (2017) in detail there are several possible reasons that sometimes the PETM-related CIE is triangular with a single large decrease in δ¹³C values, followed by an exponential recovery as seen in several marine isotope curves such as those presented in our Figure 6 and sometimes a curve, where δ¹³C values decrease in several steps. These steps can be interpreted as a result of multiple phases of carbon release. Therefore, we think that our isotope curve reflect a stepwise injection of carbon into the atmosphere comparable to the situations shown by Bowen et al. (2015) and Zhang et al. (2017). We have therefore placed the basal boundary of the CIE in our record at the least negative δ¹³C value at 3.53 m with -25.4‰, which is followed by a strong decrease to -28.1‰ at 5.37 m, only interrupted by a short but small increase of 0.4‰ at 4.57 m, which may be interpreted as carbon injection in two steps.

The position of the upper boundary of the CIE is indeed not correct in Fig. 6. Originally it should be at 16.46 m, after which a general decrease in δ¹³C values occurs. However, this decrease in δ¹³C values may have had sedimentological reasons, because the increasing δ¹³C values in the lower part of Interbed 1, which are interpreted as the recovery phase of the PETM-CIE, have been measured in light-colored silts and sands, with low content of organic material. In contrast, the sediments in the upper part of Interbed 1 from 16.46 m upwards are clayey silts with increasingly higher TOC contents (see supplementary data). A higher terrestrial organic content can be assumed here. Therefore, it is possible that the decreasing δ¹³C values are due to the stronger influence of the terrestrial carbon source. We therefore follow the reviewer's suggestion and move the upper boundary of the CIE to a depth of 22.39 m at the transition towards less negative values. According to our comment, we will discuss the upper and lower limits of the CIE in more detail in the text.

**Reviewer comment 5:** *Line 351 – here, the authors refer to the previous study by Methner et al. (2019, CP), which – according to the authors - already did suggest a position of both, the PETM and the P/E boundary below seam 1. However, when reading the study by Methner et al. (2019), one does get another impression. In this previous work, a pronounced negative anomaly (the interval entitled CIE2 in the submitted paper) is suggested to tentatively correspond to the PETM negative anomaly and comparison and correlation with PETM equivalent anomalies (Cobham lignite, Vasterival) is given. This view is now significantly revised and the part considered to correspond to the PETM by Methner et al. (2019) is now placed in the post-PETM part of the Schöningen record. This is not in itself problematic since new stratigraphic data does results in a re-interpretation of the previous data set. But this re-interpretation needs to be made clear and the statement above (line 351) should be rephrased to better represent the suggestions of Methner et al. (2019).*

**Our response:** We will rephrase the statement in line 351. Instead of "Methner et al. (2019) already suggested that both the PETM and the associated Paleocene/Eocene boundary should be placed below Seam 1." we will write:

"Methner et al. (2019) could not clearly assign the CIE between the top of Seam 1 and the lower part of Seam 2 (CIE 2 in this paper) to the PETM and already assumed that the CIE pointed to a later early Eocene thermal event. Due to a cooler mesothermal climate that is indicated by the palynoflora with mass occurrences of pollen of *Alnus*, Lenz et al. (2021) concluded that Seam 1 has been deposited during a strong temperature decline that followed the PETM in the first million years of the Eocene

(Wing et al. 2000) and suggested that both the PETM and the associated Paleocene/Eocene boundary should be placed below Seam 1."

Minor points

**Reviewer comment 6:** *Line 22: The abbreviation EECO is not explained in the abstract.*
**Our response:** We will add "Early Eocene Climatic Optimum" to explain the abbreviation.

**Reviewer comment 7:** *Line 40: Please include "to 27 to 35°C in the earliest Eocene (Inglis et al. 2020)"*
**Our response:** We will add "in the earliest Eocene".

**Reviewer comment 8:** *Line 49: The author refers to "kilo year to millennial scale". To me (and maybe to other readers) the difference between the 2 time intervals is not clear? Please explain or rephrase.*
**Our response:** We will delete "kiloyear to" in the sentence.

**Reviewer comment 9:** *Line 76 ff. There is a new published terrestrial $\delta^{13}C_{Org}$ record and associated palynology across the PETM published by Xie et al. (2022, Paleo3) entitled "Abrupt collapse of a swamp ecosystem in northeast China during the Paleocene–Eocene Thermal Maximum" which should be referred to.*
**Our response:** This is correct. Our sentence "…there are only few records providing insight into the response of terrestrial ecosystems" should include some references that use a combination of $\delta^{13}C$ data and associated palynology. Therefore, we will add Jaramillo et al. (2010) and Xie et al. (2022) as references.

**Reviewer comment 10:** *Line 272: The heading refers to the "basal Schöningen Formation" but the interval studied does represent more than half of the Schöningen Formation. Hence, the phrase "lower" might be better suited here…*
**Our response:** We will change "basal Schöningen Formation" in "lower Schöningen Formation"

**Reviewer comment 11:** *Line 421: Change to "observed in the terrestrial records".*
**Our response:** We will change this part of the sentence as suggested.

References:
Bowen, G., Maibauer, B., Kraus, M., Röhl, U., Westerhold, T., Steimke, A., Gingerich, P. D., Wing, S. L., and Clyde, W. C.: Two massive, rapid releases of carbon during the onset of the Palaeocene–Eocene thermal maximum, Nature Geoscience, 8, 44–47, https://doi.org/10.1038/ngeo2316, 2015.

Collinson, M. E., Hooker, J. J., and Gröcke, D. R.: Cobham Lignite Bed and penecontemporaneous macrofloras of southern England: A record of vegetation and fire across the Paleocene-Eocene Thermal Maximum, in: Causes and consequences of globally warm climates in the early Paleogene, edited by: Wing, S. L., Gingerich, P. D., Schmitz, B., and Thomas, E., GSA Special Paper, 369, 333–349, https://doi.org/10.1130/0-8137-2369-8.333, 2003.

Diefendorf, A. F., Mueller, K. E., Wing, S. L., Koch, P. L., and Freeman, K. H.: Global patterns in leaf 13C discrimination and implications for studies of past and future climate, PNAS, 107, 5738-5743; https://doi.org/10.1073/pnas.0910513107, 2010.

Domingo, L., López-Martínez, N., Leng, M. J., and Grimes, S. T.: The Paleocene–Eocene Thermal Maximum record in the organic matter of the Claret and Tendruy continental sections (South-central Pyrenees, Lleida, Spain), Earth and Planetary Science Letters 281, 226–237, https://doi.org/10.1016/j.epsl.2009.02.025, 2009

Jaramillo, C., Ochoa, D., Contreras, L., Pagani, M., Carvajal-Ortiz, H., Pratt, L. M., Krishnan, S., Cardona, A., Romero, M., Quiroz, L., Rodriguez, G., Rueda, M. J., de la Parra, F., Morón, S., Green, W., Bayona, G., Montes, C., Quintero, O., Ramirez, R., Mora, G., Schouten, S., Bermudez, H., Navarewtte, R., Parra, F., Alvarán, M., Osorno, J., Crowley, J. L., Valencia, V., and Vervoort, J.:

Effects of rapid global warming at the Paleocene-Eocene Boundary on Neotropical Vegetation, Science, 330, 957–961, https://doi.org/10.1126/science.1193833, 2010.

Lenz, O. K, Riegel, W., and Wilde V.: Greenhouse conditions in lower Eocene coastal wetlands? - Lessons from Schöningen, Northern Germany, PLoS ONE, 16, e0232861, https://doi.org/10.1371/journal.pone.0232861, 2021.

Methner, K., Lenz, O. K., Riegel, W., Wilde, V., and Mulch, A.: Paleoenvironmental response of midlatitudinal wetlands to Paleocene–early Eocene climate change (Schöningen lignite deposits, Germany), Clim. Past, 15, 1741–1755, https://doi.org/10.5194/cp-15-1741-2019, 2019.

Schouten, S., Woltering, M., Rijpstra, W. I. C., Sluijs, A., Brinkhuis, H., and Sinninghe Damsté, J. S.: The Paleocene–Eocene carbon isotope excursion in higher plant organic matter: Differential fractionation of angiosperms and conifers in the Arctic, Earth and Planetary Science Letters, 258, 581–592, https://doi.org/10.1016/j.epsl.2007.04.024, 2007.

Sluijs, A. and Dickens, G. R.: Assessing offsets between the $\delta13C$ of sedimentary components and the global exogenic carbon pool across early Paleogene carbon cycle perturbations. Global Biogeochem. Cycles, 26, GB4005, https://doi.org/10.1029/2011GB004224, 2012.

Sluijs, A., van Roij, L., Frieling, J., Laks, J., and Reichert, G.-J.: Single-species dinoflagellate cyst carbon isotope ecology across the Paleocene-Eocene Thermal Maximum, Geology, 46, 79–82, https://doi.org/10.1130/G39598.1, 2018.

Smith, F. A., Wing, S. L., and Freeman, K. H.: Magnitude of the carbon isotope excursion at the Paleocene-Eocene thermal maximum: the role of plant community change, Earth Planet. Sc. Lett., 262, 50–65, https://doi.org/10.1016/j.epsl.2007.07.021, 2007.

Steurbaut, E., Magioncalda, R., Dupuis, C., Van Simaeys, S., Roche, E., and Roche, M.: Palynology, paleoenvironments, and organic carbon isotope evolution in lagoonal Paleocene-Eocene boundary settings in North Belgium, in: Causes and consequences of globally warm climates in the early Paleogene, edited by: Wing, S. L., Gingerich, P. D., Schmitz, B., and Thomas, E., GSA Special Paper, 369, 291–317, https://doi.org/10.1130/0-8137-2369-8.291, 2003.

Storme, J. Y., Dupuis, C., Schnyder, J., Quesnel, F., Garel, S., Iakovleva, A. I., Iacumin, P., Di Matteo, A., Sebilo, M., and Yans, J.: Cycles of humid-dry climate conditions around the P/E boundary: new stable isotope data from terrestrial organic matter in Vasterival section (NW France), Terra Nova, 24, 114–122, https://doi.org/10.1111/j.1365-3121.2011.01044.x, 2012.

Tremblin, M., Khozyem, H., Adatte, T., Spangenberg, J. E., Fillon, C., Grauls, A., Hunger, T., Nowak, A., Läuchli, C., Lasseur, E., Roig, J.-Y., Serrano, O., Calassou, S., Guillocheau, F., and Castelltort, S.: Mercury enrichments of the Pyrenean foreland basins sediments support enhanced volcanism during the Paleocene-Eocene thermal maximum (PETM), Global and Planetary Change, 212, 103794, https://doi.org/10.1016/j.gloplacha.2022.103794, 2022.

Xie, Y., Wu, F., and Fang, X.: Abrupt collapse of a swamp ecosystem in northeast China during the Paleocene–Eocene Thermal Maximum, Palaeogeography, Palaeoclimatology, Palaeoecology, https://doi.org/10.1016/j.palaeo.2022.110975, 2022.

Zhang, Q., Wendler, I., Xu, X., Willems, H., and Ding, L.: Structure and magnitude of the carbon isotope excursion during the Paleocene-Eocene thermal maximum, Gondwana Research 46, 114–123, http://dx.doi.org/10.1016/j.gr.2017.02.016, 2017.